# AN INVESTIGATION OF REPRESENTATION AND ALLOCATION HARMS IN CONTRASTIVE LEARNING

**Subha Maity**[*]
Department of Statistics
University of Michigan
Ann Arbor, MI
smaity@umich.edu

**Mayank Agarwal**
IBM Research
MIT-IBM Watson Lab
Cambridge, MA
mayank.agarwal@ibm.com

**Mikhail Yurochkin**
IBM Research
MIT-IBM Watson Lab
Cambridge, MA
mikhail.yurochkin@ibm.com

**Yuekai Sun**
Department of Statistics
University of Michigan
Ann Arbor, MI
yuekai@umich.edu

## ABSTRACT

The effect of underrepresentation on the performance of minority groups is known to be a serious problem in supervised learning settings; however, it has been underexplored so far in the context of self-supervised learning (SSL). In this paper, we demonstrate that contrastive learning (CL), a popular variant of SSL, tends to collapse representations of minority groups with certain majority groups. We refer to this phenomenon as representation harm and demonstrate it on image and text datasets using the corresponding popular CL methods. Furthermore, our causal mediation analysis of allocation harm on a downstream classification task reveals that representation harm is partly responsible for it, thus emphasizing the importance of studying and mitigating representation harm. Finally, we provide a theoretical explanation for representation harm using a stochastic block model that leads to a representational neural collapse in a contrastive learning setting.

## 1 INTRODUCTION

Most prior studies of algorithmic bias focus on *allocation harms*: objectionable demographic disparities in the allocation of resources (Angwin et al., 2016; O'Neil, 2017; Dastin, 2018). They are widely studied because they are easy to measure (simply measure disparities in resource allocation), and they can be tied to common performance metrics in supervised learning (*e.g.* misclassification rates). For example, consider a credit default risk assessment model that classifies potential borrowers as high-risk and low-risk. Any disparities in the misclassification rate between borrowers from different demographic groups lead to disparities in lending rates. This focus on allocation harms is reflected in the plethora of methods for mitigating such harms, including methods based on resampling and reweighting (He et al., 2008; Ando & Huang, 2017; Byrd & Lipton, 2019), enforcing invariance constraints (Agarwal et al., 2018), re-calibrating the logits (Tian et al., 2020).

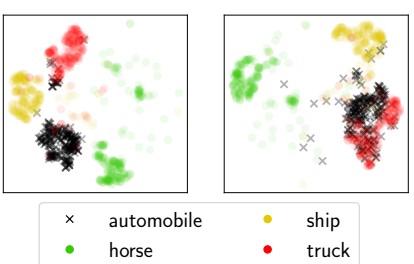

Figure 1: tSNE visualization of CL representations for balanced (left) and `automobile` under-represented (right) CIFAR10 data.

In contrast to the voluminous literature on allocative harms, representation harms have received less attention. This is because representation harms are harder to measure and their effects are

---

[*]Corresponding author. Accompanying codes can be found in https://github.com/smaityumich/CL-representation-harm.

more diffuse and longer-term (O'Neil, 2017). For example, Google Photos mistakenly tags black people as gorillas (Barr, 2015). There are no resources that are being misallocated here; rather, the harm is inflicted by perpetuating historical biases against black people. Despite recent advances in representation learning (Bengio et al., 2013), there is a lack of research on algorithmic biases in this area, especially studies that look into representation harms they may cause. In this paper, we seek to fill this gap in the literature by investigating the effects of group underrepresentation in contrastive learning (CL) algorithms (Chen et al., 2020; Chen & He, 2021; He et al., 2020). CL is a popular instance of self-supervised learning (SSL), an approach for learning representations that can effectively leverage unlabeled samples to improve (downstream) model performances (Wang et al., 2021; Babu et al., 2021). However, these datasets often exhibit inherent imbalances (De-Arteaga et al., 2019) and it is important to know whether training a CL model on such datasets may detrimentally affect the quality of representations and cause downstream allocative and representation harms.

Here is a preview of the results of our controlled study of underrepresentation with the CIFAR10 dataset. In Figure 1 we plot 2D t-SNE embeddings (Van der Maaten & Hinton, 2008) of CL representations of vehicles in the dataset. We see that underrepresentation of `automobile` images in CL training causes their representation to cluster with those of `trucks`. This is an instance of stereotyping (Abbasi et al., 2019) because samples from an underrepresented group are lumped together with those of a similar majority group; thus erasing the unique characteristics of the underrepresented. This is a form of representation harm (Crawford, 2017), which can lead to downstream allocation harms (*e.g.* missclassifications between `automobiles` and `trucks`). In this paper, we show that this mechanism of representation to downstream allocation harms is general. This complements prior works, which show that underrepresentation leads to allocation harms without identifying the underlying mechanism. The rest of the paper is organized as follows.

1. In Section 2 we empirically show that underrepresentation leads to representation harms, as the CL representations of underrepresented groups collapse to semantically similar groups.

2. In Section 3 we develop a simple model of CL on graphs that exhibits a collapse of (the learned representations of) underrepresented groups to (those of) semantically similar groups. This suggests that the representation harms we measured are intrinsic to CL representations.

3. In Section 4 we show via a (causal) mediation analysis that some of the allocation harms from a linear head/probe (trained on top of the CL representations) can be attributed to the representation harms in the CL representations. Thus, in order to mitigate the allocation harms from ML models built on top of CL representations, we must mitigate the harms in the learned representations.

## 1.1 RELATED WORKS

The paradigm of **self-supervised learning (SSL)** allows the use of large-scale unsupervised datasets to train useful representations and has drawn significant attention in modern machine learning (ML) research (Misra & Maaten, 2020; Jing & Tian, 2020). It has found applications in a broad spectrum of areas, such as computer vision (Chen et al., 2020; He et al., 2020), natural language processing (Fang et al., 2020; Liu et al., 2021a; Hsu et al., 2021), and many more. In this paper, we investigate the effect of underrepresentation in contrastive learning (CL) (Chen et al., 2020; Chen & He, 2021; He et al., 2020), which is a popular variant for SSL, that uses similar samples to learn representations. A more extensive discussion of previous work can be found in the surveys Schiappa et al. (2023); Yu et al. (2023); Liu et al. (2022) for SSL and Le-Khac et al. (2020); Kumar et al. (2022) for CL. In this paper, we use SimCLR (Chen et al., 2020) and SimSiam (Chen & He, 2021) for CIFAR10 dataset and SimCSE (Gao et al., 2021) for BIASBIOS dataset to learn CL representations.

Several works have **theoretically studied** the success of **self-supervised learning** (Arora et al., 2019; HaoChen et al., 2021; Lee et al., 2020; Tian et al., 2021; Tosh et al., 2021). Our theoretical analysis of CL loss is partly motivated by Fang et al. (2021b), who showed that CL representations of a group collapse to a single vector. This phenomenon is known as **neural collapse**, and it also occurs in supervised learning (Papyan et al., 2020; Zhu et al., 2021; Fang et al., 2021a;b; Hui et al., 2022; Han et al., 2021) and transfer learning (Galanti et al., 2021). Our analysis differs from Fang et al. (2021b) because they study neural collapse when the dataset is balanced, while we focus on imbalanced datasets. This is crucial for studying allocation and representation harms caused by underrepresentation. We show that when the dataset is imbalanced, neural collapse still occurs, but the collapsed points no longer possess a symmetric simplex structure. Instead, the relative positions of

the collapsed points depend on the degree of underrepresentation in an intricate manner; we explore the resulting structure in Section 3.2.

Liu et al. (2021c) showed that under group imbalance, predictive models based on SSL representations cause less allocation harm than models trained in a supervised end-to-end manner. However, their work does not preclude allocation harms by SSL. We build on their work by studying the allocation and representation harms caused by SSL.

Previous work on mitigating representation harm is discussed in Section 5.

## 2   UNDER-REPRESENTATION CAUSES STEREOTYPING

In Figure 1 we observe that underrepresentation of `automobiles` causes its CL representations to collapse with `trucks`. To delve deeper into this phenomenon, in this section, we consider two cases of underrepresentation: (i) a controlled study with CIFAR10 dataset, where we emulate the effect of underrepresentation by systemically subsampling a class, and (ii) on BIASBIOS dataset, which is naturally imbalanced.

### 2.1   CONTROLLED STUDY OF CLASS UNDERREPRESENTATION

#### 2.1.1   EXPERIMENTAL SETUP

**Dataset:**   For our controlled study, we consider CIFAR10 (Krizhevsky et al., 2009), a well-known visual benchmark dataset that has 10 classes (`airplane`, `cars`, etc.) and both the training and the test datasets are balanced (equal number of images in all classes). To simulate underrepresentation, we randomly subsample 1% of the images for one of the classes when training our CL models. We denote this dataset as $\mathcal{D}_k$, where $k$ is the class that is undersampled. Furthermore, we train a CL model on the balanced dataset ($\mathcal{D}_\text{bal}$) as a baseline.

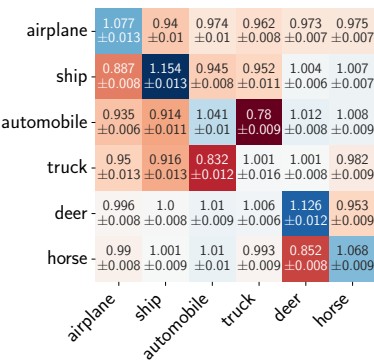

Figure 2: Representation harms in CIFAR10 over 10 repetitions.

**Models and representations:**   We use the ResNet-34 backbone and train on both balanced and underrepresented training datasets using two well-known CL approaches: SimCLR (Chen et al., 2020) and SimSiam (Chen & He, 2021). We will refer to the model trained with a balanced (resp. underrepresented) dataset as a balanced (resp. underrepresented) model. In the main text, we present results for SimCLR. The results for SimSiam are deferred to Appendix A.2 and are quite similar to the SimCLR results. Further details and results of the SimCLR implementation can be found in Appendix A.1.

#### 2.1.2   REPRESENTATION HARM

In Figure 1 we see that the underrepresentation in `automobile` class causes its CL representations to merge with those of `trucks`, erasing the distinction between automobiles and trucks. We suspect a similar phenomenon occurred when Google mistakenly tagged black people are Gorillas: the embeddings of black people merged with those of Gorillas, erasing the distinction between the two classes. Here, we further investigate this behavior, *i.e.*, whether this phenomenon persists among other semantically similar pairs.

**Metric:**   We measure the representation harm between a pair of classes $(l, m)$ when the $k$-th class is underrepresented as

$$\text{RH}(l, m; k) = \frac{\mathbf{D}(l,m;f_k)}{\mathbf{D}(l,m;f_\text{bal})}, \quad \mathbf{D}(l, m; f_k) = \frac{1}{n_l n_m} \sum_{\substack{i \in [n_l], \\ j \in [n_m]}} \left\{ 1 - \cos\left(f_k(x_{l,i}), f_k(x_{m,j})\right) \right\}, \quad (2.1)$$

where $f_\text{bal}$ (resp. $f_k$) is the balanced (resp. underrepresented) CL model trained on $\mathcal{D}_\text{bal}$ (resp. $\mathcal{D}_k$). $\text{RH}(l, m; l) < 1$ implies that underrepresentation in class $l$ causes its CL representation to collapse with class $m$. In Figure 2 we plot the representation harm metrics, where the $(l, m)$-th entry is $\text{RH}(l, m; l)$. We present RH metrics for the four vehical classes (`airplane`, `ship`, `automoble`

and `truck`), and two animal classes (`deer` and `horse`). The other RH metrics can be found in Appendix A.1.1.

**Results:** In Figure 2 we observe the worst case of representation harm between `automobile` and `truck` classes. The RH metric for this pair is the lowest ($0.78 \pm 0.009$) when `automobile` is undersampled, and second lowest ($0.832 \pm 0.012$) when `truck` is undersampled. This means underrepresentation in either `automobile` or `truck` results in a collapse in their CL representations with each other. Similar behavior is also observed between pairs (`airplane`, `ships`) (RH metrics are $0.887 \pm 0.008$ and $0.94 \pm 0.01$) and (`deers`, `horses`) (RH metrics are $0.852 \pm 0.008$ and $0.953 \pm 0.009$). Since (`automobiles`, `trucks`) are vehicles on road , (`ships`, `airplanes`) are vehicles with blue (water/sky) backgrounds, and (`deers`, `horses`) are mammals with green backgrounds, these pairs are semantically similar pairs, which affirms our intuition that CL representations of an underrepresented class collapse with the representations of a similar class. We shall see later in Section 4.2 that these representation harms cause allocation harm, *i.e.* reduction in downstream classification accuracies.

## 2.2 EFFECTS OF UNDER-REPRESENTATION IN THE WILD

An important premise of self-supervised learning is the ability to train on large amounts of data from the Internet with little or no data curation. Such data will inevitably contain many underrepresented groups. In this experiment, we study the potential harms of CL applied to data obtained from Common Crawl, a popular source of text data for self-supervised learning.

### 2.2.1 EXPERIMENTAL SETUP

**Dataset:** We consider BIASBIOS dataset (De-Arteaga et al., 2019) which consists of around 400k online biographies in English extracted from the Common Crawl data. These biographies represent 28 occupations appearing at different frequencies (`professor` being the most common, and `rapper` the least common). In addition, many occupations are dominated by biographies of either male or female gender (identified using the pronouns in the biographies), mimicking societal gender stereotypes. Please see Appendix B for additional details.

**Model and representations:** We obtain CL representations by fine-tuning BERT (Devlin et al., 2018) with SimCSE loss (Gao et al., 2021), an analog of SimCLR for text data, that uses dropout in the representation space instead of image augmentations. We use a random 75% of samples for training the SimCSE representations and the remaining data for computing all metrics reported in the experiments. SimCSE representations are trained using the source code from the authors in the unsupervised mode and with the MLP projection layer (Gao et al., 2021).

### 2.2.2 REPRESENTATION HARM

As discussed previously, we expect the representations of underrepresented groups to collapse towards the groups that are most similar to them. For example, consider the class `surgeon` which consists of 85% male and 15% female biographies. Will the biographies of underrepresented female `surgeons` be assigned representations similar to male `surgeons` or to female biographies of a different but related occupation such as `dentist`? Although none of these outcomes are desirable, the latter can cause representation harms associated with gender stereotyping.

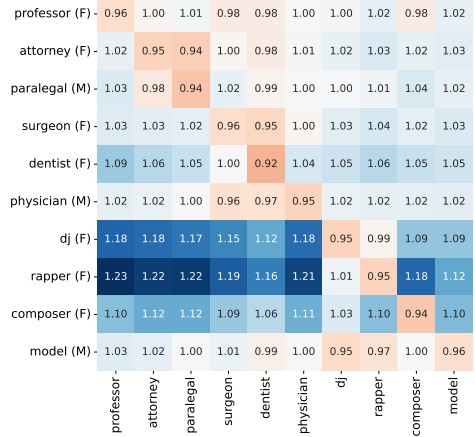

Figure 3: Gender RH in BIASBIOS dataset.

**Metric:** To measure the representation harm and answer the aforementioned question we consider a variant of the metric based on the average cosine distance used in the CIFAR10 experiment. Let $\mathbf{D}(l_g, m_{g'})$ be the average cosine distance between the representations of gender $g$ biographies from occupation $l$ and gender $g'$ biographies from occupation $m$ (analogous to equation 2.1 where we omit the dependency on the model $f$ since we have a single CL model in this experiment). Let $g$

be the underrepresented gender for occupation $l$, we define gender representation harm (GRH) for occupations $l$ and $m$ as:

$$\text{GRH}(l, m) = \frac{\mathbf{D}(l_g, m_g)}{\mathbf{D}(l_g, l_{g'})}. \tag{2.2}$$

$\text{GRH}(l, m) < 1$ implies that the learned representations for the underrepresented gender in occupation $l$ are *closer* to representations of occupations $m$ for the same gender than they are to the representations for the different gender within the same occupation. $\text{GRH}(l, m) < 1$ for $l \neq m$ can be interpreted as a warning sign of gender stereotyping in representations.

**Results:**  We present the GRH results for a subset of occupations in Figure 3 (F and M in the row names indicate the underrepresented gender for the corresponding occupation; complete results are in Appendix B.3). We focus on the off-diagonal entries which should be >1 when there is no representation harm. We observe several deviations from this rule, especially for occupation pairs (`attorney`, `paralegal`) and (`surgeon`, `dentist`). Specifically, GRH(`attorney`, `paralegal`) of 0.94 suggests that representations of female attorneys are closer to representations of female paralegals than they are to that of male attorneys. The two occupations are highly related and such representation harm can result in disadvantaging female attorneys. Analogously, GRH(`surgeon`, `dentist`) of 0.95 corresponds to a similar problem for a pair of occupations in the medical domain.

## 3 ASYMPTOTIC ANALYSIS OF CL REPRESENTATIONS

In this section, we show theoretically that the representation harms in contrastively learned representations are generic and are not specific to certain datasets. Here, we focus on a generic contrastive learning loss (Chen et al., 2020; Wang & Isola, 2020):

$$\min_{\Phi: \mathcal{X} \to \mathbf{S}^{d-1}} \mathbf{L}_{\text{CL}}(\Phi(X)),$$
$$\mathbf{L}_{\text{CL}}(V) \triangleq -\frac{1}{n} \sum_{i \in [n]} \frac{1}{\sum_{j \in [n]} e(i,j)} \sum_{j \in [n]} e(i,j) \log \left\{ \frac{\exp(v_i^\top v_j / \tau)}{\frac{1}{n} \sum_{l \in [n]} \exp(v_i^\top v_l / \tau)} \right\}, \tag{3.1}$$

where $\mathbf{S}^{d-1}$ is the unit sphere in $\mathbb{R}^d$, the rows of $\Phi(X) \in \mathbb{R}^{n \times d}$ are the learned representations of the samples $X_i$, and $v_i \in \mathbb{R}^d$ is the (learned) representation of the $i$-th sample. Note that the loss depends on the raw inputs $X_i$'s only through their similarities $e(i, j)$. We shall exploit this fact to simplify the subsequent theoretical developments. In practice, $\Phi$ is usually parameterized as a neural network.

To isolate the effects of the loss (from other inductive biases encoded in the architecture of $\Phi$ and the training algorithm), we make the layer-peeled assumption (Fang et al., 2021a;b) that $\Phi$ can produce any point in $\mathbf{S}^{d-1}$. This assumption simplifies the problem to optimizing directly on the outputs of $\Phi$ (instead of the parameters of $\Phi$).

$$\min_{v_i \in \mathbf{S}^{d-1}} \mathbf{L}_{\text{CL}}(V), \tag{3.2}$$

where the optimizers $v_i^\star$ corresponds to the learned representation of $X_i$. Since the CL loss depends only on the similarities between samples, we directly impose a probabilistic model on the similarities (instead of imposing assumptions on the distribution of samples). This simplifies the theoretical developments.

**Stochastic block model:**  We formalize similarities between samples as a similarity graph on the samples and impose a stochastic block model (SBM) (Holland et al., 1983) on the similarity graph. Typically, an SBM with $K$ blocks is described by two parameters: (1) $\Pi = [\pi_1, \ldots, \pi_K]^\top$ the probabilities of a data point or node belonging to each of the blocks, and (2) $A = [[\alpha_{k,k'}]]_{k,k' \in [K]}$, a matrix that describes connectivity between blocks, where $\alpha_{k,k'}$ is the probability that a node from block $k$ is connected to a node from block $k'$. The $\pi_k$'s controls the size of $k$-th block and $\alpha_{k,k'}$'s determines the similarity between blocks. A sample of size $n$ is drawn from SBM($\Pi, A$) as:

1. We generate $n$ nodes from a multinomial distribution with probability vector $\Pi$. Let us denote $\{Y_i\}_{i=1}^n \subset [K]$ as the block annotations, *i.e.* the $i$-th node belongs to the $Y_i$-th block. Note that $Y_i$ are i.i.d. categorical($\Pi$).

2. For each pair of nodes, we generate $e(i, i') \mid Y_i, Y_{i'} \sim \text{Bernoulli}(\alpha_{Y_i, Y_{i'}})$, where $e(i, i')$ indicates whether the nodes $i$ and $i'$ are connected by an undirected edge. In the context of CL from images, such edges imply that neighboring nodes $i$ and $i'$ are augmentations of the same or similar images.

The SBM has two desirable properties:

1. An inherent *group structure* that is often associated with downstream classification tasks. For example, the classes in CIFAR10 dataset can be considered as groups.

2. A notion of *connectivity* among groups that explains which of them are similar. As we shall see later, this is a key factor in identifying the majority groups to which representations of an underrepresented group can collapse. Recall, in the example of CIFAR10 (Section 4) we see that the `automobile` and `truck` classes are connected to each other, and underrepresentation of either of them results in the collapse of their CL representations.

### 3.1 SIMULATIONS WITH THE STOCHASTIC BLOCK MODEL

To understand how underrepresentation may cause representation harm in CL representations, we conduct a simple experiment on SBM data. We consider an SBM with three blocks. The connectivity probabilities are provided in the left plot of Figure 4, where only the first two groups are connected with each other. Additional details are provided in Appendix D.

In our experiment, we underrepresent the first block by factors of $2^{-2}$ and $2^{-4}$ (the sample size for the first group is reduced from $2^6$ to $2^4$ and $2^2$) and investigate its representation harm. For this purpose, we obtained

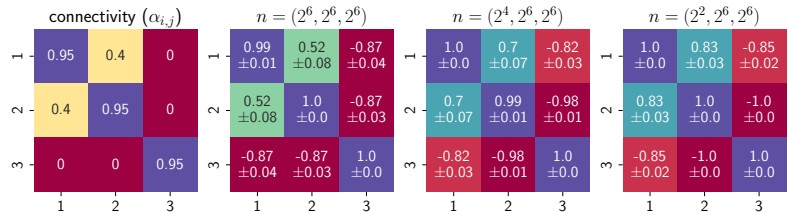

Figure 4: *Left:* connectivity across groups, *middle-left, middle-right and right:* average cosines and their standard deviations across groups.

the CL representations from equation 3.2. Note that the generic CL cost function is exactly the popular node2vec cost function (Grover & Leskovec, 2016) for graph representation learning, so our results also have implications in graph representation learning. In Figure 4 we present the means and standard deviations for the cosines of the CL representations between groups. Some notable observations are described below.

- The diagonals in Figure 4 demonstrate that all $v_i^\star$ vectors within a block collapse into a single vector, as they have cosines very close to one. This phenomenon is known as *neural collapse* (Papyan et al., 2020) and has been observed by Fang et al. (2021b) in analyzing CL representation with balanced data. Further related discussions are deferred to Section 3.2.

- The same plots suggest that the CL representations of the first two groups get closer when the first group is underrepresented, as seen in their increased cosine similarity (it increases from $0.52(\pm0.08)$ to $0.7(\pm0.07)$ in the middle right and $0.83(\pm0.03)$ in the right plot). This is a form of representation harm, as the representations of the first two groups, which are similar to each other, partially collapse in terms of their CL representations (RH metric in equation 2.1 are $RH(1,2;1) = 0.48$ and $0.35$). Additionally, the second and third groups become further apart as their cosines decrease from $-0.87(\pm0.03)$ to $-0.99(\pm0.0)$ and $-1(\pm0.0)$.

### 3.2 ASYMPTOTIC ANALYSIS OF CL REPRESENTATIONS FOR STOCHASTIC BLOCK MODELS

Here, we verify the two phenomena that we observe in our simulation with the SBM dataset: (1) the collapse of CL representations within a block and (2) the representation harm between blocks with high connectivity. For this purpose, we perform an asymptotic analysis on the optimal CL representations $v_i^\star$ at $n$ going to infinity. We require the following assumption in our analysis.

**Assumption 3.1.** *Denoting the representation variables of the $k$-th block as $\{v_{k,j}\}_{j=1}^{n_k}$, where $n_k = \#\{Y_i = k\}$ and for each $k$ we assume that $1/n_k \sum_{j=1}^{n_k} v_{k,j}$ converges as $n_k \to \infty$.*

The above assumption simply requires that the representation variables in the optimization of equation 3.2 that belong to the same block do not fluctuate too much, which is a rather minimal assumption. With this assumption, we state our main result.

**Theorem 3.2.** *Assume that $\sum_{k' \in [K]} \pi_{k'} \alpha_{k,k'} > 0$ for each $k$. Then at $n \to \infty$ the optimum CL representations obtained from the minimization of problem in equation 3.2 satisfy the following: $v_i^\star \overset{a.s.}{=} h_{Y_i}^\star$, where $\overset{a.s.}{=}$ denotes almost sure equality and $\{h_k^\star\}_{k \in [K]}$ is a minimizer for*

$$\min_{h_k \in \mathbf{S}^{d-1}} - \sum_{k_1=1}^{K} \pi_{k_1} \frac{\sum_{k_2=1}^{K} \pi_{k_2} \alpha_{k_1,k_2} (h_{k_1}^\top h_{k_2}/\tau)}{\sum_{k_2=1}^{K} \pi_{k_2} \alpha_{k_1,k_2}} + \sum_{k_1=1}^{K} \pi_{k_1} \log \left\{ \sum_{k_3=1}^{K} \pi_{k_3} e^{h_{k_1}^\top h_{k_3}/\tau} \right\}. \quad (3.3)$$

*Note that the objective in equation 3.3 is a weighted version of the generic CL objective in equation 3.2 applied to common group-wise representations $h_{Y_i}$.*

This theorem shows that all samples from the same block have the same representation as $n \to \infty$; *i.e.* all $v_i^\star$'s converge to their corresponding $h_{Y_i}^\star$s. This phenomenon is known as *neural collapse*. It has been studied in the context of layer peeled model for supervised learning (Papyan et al., 2020; Zhu et al., 2021; Mixon et al., 2020; Lu & Steinerberger, 2020), and in CL representations with balanced data (Fang et al., 2021b). Fang et al. (2021b)'s results are most similar to ours, but they do not elicit the effects of dataset imbalance on the geometry of the collapsed representations.

**Representation harm:** To understand the representation harm we reconsider the setup of the synthetic experiment in Section 3.1. The SBM has three blocks, where for a given $\pi_1$ we set $\pi_2 = \pi_3 = (1 - \pi_1)/2$. In the leftmost plot of Figure 5 we present the connectivity matrix, and as before, only the first two blocks are

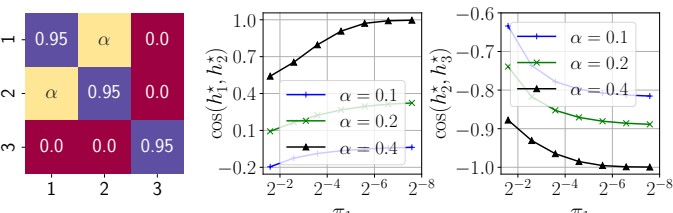

Figure 5: *Left:* connectivity, and *middle and right:* cosine similarities between groups $(1, 2)$ and $(2, 3)$.

connected with probability $\alpha$. In this setup, we obtain representations $h_k^\star$ from equation 3.3 and plot their cosine similarities for pairs $(1, 2)$ and $(2, 3)$ in Figure 5. The harm in representation becomes more prominent between the first two groups with the severity of the underrepresentation of the minority group, as their cosine increases in the middle plot of Figure 5. In fact, their representations become identical for $\pi_1 < 2^{-6}$ when there is sufficient connectivity between the first two groups ($\alpha = 0.4$). Additionally, the two disconnected majority groups (second and third groups) get further apart, and at the extreme ($\pi_1 > 2^{-6}$) they become exactly opposite (cosine is $-1$).

Finally, when the first two groups are less connected ($\alpha = 0.1$), their representations do not collapse, even for a severe underrepresentation of the first group (cosine is less than zero when $\pi_1 < 2^{-6}$). This relates to an observation in CIFAR10 case-study in Figure 8a in Appendix A.1.1, that the `dog` class, which is arguably disconnected from other classes, suffers the least allocation harm when it is underrepresented.

**Practical implications:** Reiterating our previous discussions, our analysis suggests that harm in representation occurs between two semantically similar groups when either of them is underrepresented. Therefore, to mitigate the issue, practitioners should consider a combination of the following strategies; 1) reweighing the loss to counteract underrepresentation, an approach considered in Liu et al. (2021c); Zhou et al. (2022), and 2) a "surgery" on connectivity, similar to the approach in Ma et al. (2021). We defer our further discussion of the mitigation of harm to Section 5.

## 4 HARMS IN CL REPRESENTATIONS CAUSE ALLOCATION HARMS

In Section 2 we observed that underrepresentation of a group causes its CL representation to collapse with symantically similar groups. Through a causal mediation analysis (CMA) (Pearl, 2022), in this section, we show that this is partly responsible for allocation harm (AH) in downstream classification tasks. Thus, to fully mitigate downstream allocation harm, practitioners must address this representation harm.

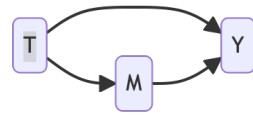

Figure 6: Basic model for causal mediation analysis

## 4.1 BACKGROUND ON CAUSAL MEDIATION ANALYSIS

The goal of CMA is to decompose the causal effect of a treatment on an outcome into a direct (causal) effect that goes directly from the treatment to the outcome and an indirect effect that goes through other variables in the causal graph. In this section, we consider the basic CMA graph shown in Figure 6. Here, treatment $T$ is whether the dataset is undersampled, $M$ is the CL representations, and $Y$ is a measure of downstream allocation harm (*e.g.* misclassification rate). To introduce our decomposition we require the following notation. The outcome under treatement $T = t$ is denoted as $Y_{\text{do}(T=t)}$, and the same under treatment $T = t$ and mediation $M = m$ is denoted as $Y_{\text{do}(T=t,M=m)}$. Similarly, mediation under treatment $t$ is denoted as $M_t = M_{\text{do}(T=t)}$. Note that $Y_{\text{do}(T=t)} = Y_{\text{do}(T=t,M=M_t)}$.

The two effects that we evaluate are the **natural indirect effect (NIE)** and the **reverse natural direct effect (rNDE)**. It is easily seen that they add up to the total effect $\text{TE} \triangleq \mathbb{E}\big[Y_{\text{do}(T=1)}\big] - \mathbb{E}\big[Y_{\text{do}(T=0)}\big]$:

$$\text{TE} = \underbrace{\mathbb{E}\big[Y_{\text{do}(T=1,M=M_1)}\big] - \mathbb{E}\big[Y_{\text{do}(T=0,M=M_1)}\big]}_{-\text{rNDE}} + \underbrace{\mathbb{E}\big[Y_{\text{do}(T=0,M=M_1)}\big] - \mathbb{E}\big[Y_{\text{do}(T=0,M=M_0)}\big]}_{\text{NIE}}$$

(4.1)

Here, the NIE is the downstream allocation that can be attributed to (representation harms) in the CL representation, while -rNDE is the downstream allocation harms directly caused by undersampling.

## 4.2 MEDIATION ANALYSIS ON CONTROLLED STUDY

In our controlled study with CIFAR10 dataset in Section 2.1 we observed that semantically similar pairs such as (`automobiles`, `trucks`), (`airplanes`, `ships`), and (`deers`, `horses`) suffer from representation harm when one of the classes is underrepresented. Following this, using mediation analysis, we show that it may cause allocation harm in a downstream classification, thus emphasizing its importance in mitigating allocation harm.

**Setup:** Suppose that treatment $T = 1$ is the underrepresentation of class $k$. We train a linear head with a randomly chosen 75% of the test data, considering two scenarios: (1) it is trained on a balanced dataset ($T = 0$), and (2) it is trained on an imbalanced dataset where the class $k$ is subsampled to 1% of its original size ($T = 1$). Recalling that $f_{\text{bal}}$ (resp. $f_k$) denotes the CL model trained on balanced (resp. class $k$ underrepresented) training dataset, we denote $f_{\text{bal}}$ as $M_{\text{do}(T=0)}$ and $f_k$ as $M_{\text{do}(T=1)}$. A CL model coupled with

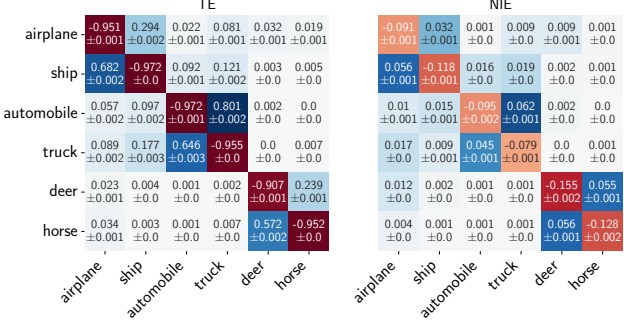

Figure 7: TE (*left*), −rNDE (*middle*) and NIE (*right*) for CIFAR10 dataset over 10 repetitions.

a linear head builds the final image classifier, which we always evaluate on the remaining 25% of the test dataset.

**Metrics:** We consider three classifiers, where a linear head is trained on top of: (1) $f_{\text{bal}}$ using balanced data ($T = 0, M = M_{\text{do}(T=0)}$), which we denote as $\hat{y}_{0,0}$, (2) $f_k$ using balanced data ($T = 0, M = M_{\text{do}(T=1)}$), denoted as $\hat{y}_{0,1}$, and finally (3) $f_k$ using imbalanced data ($T = 1, M = M_{\text{do}(T=1)}$), denoted as $\hat{y}_{1,1}$. With all the notations in hand, the total effect (resp. natural indirect effect) due to the underrepresentation of the class $k$ in classifying a sample $x$ with the true class $y = k$ as $\hat{y} = l$ is

$$\text{TE}(k,l) = P(\hat{y}_{1,1}^{(k)}(x) = l \mid y = k) - P(\hat{y}_{0,0}(x) = l \mid y = k)$$
$$\text{NIE}(k,l) = P(\hat{y}_{0,1}^{(k)}(x) = l \mid y = k) - P(\hat{y}_{0,0}(x) = l \mid y = k).$$

(4.2)

Finally, the corresponding reverse natural direct effect is calculated as $-\text{rNDE}(k,l) = \text{TE}(k,l) - \text{NIE}(k,l)$. For $k \neq l$ the $\text{TE}(k,l) > 0$ indicates how much more often the class $k$ is mistaken for the class $l$ when $k$ is underrepresented and $\text{NIE}(k,l)$ is the part that is caused by representation harm.

We present the TE, $-$rNDE and NIE as heatmap plots in Figure 7, where TE$(k, l)$ is the entry of the $(k, l)$-th cell and similarly for $-$rNDE and NIE. As in Section 2.1.2, we present the metrics for only six classes (four vehicles and two animals). The remaining ones are provided in Appendix A.1.1.

**Results:** In Figure 7 we mainly focus on the NIE, as this part of allocation harm caused by harm in CL representation. When underrepresented, `automobiles` are mistaken as `truck` most often (off-diagonal TE is highest at $0.798 \pm 0.002$). Additionally, the allocation harm caused by harm in CL representations is the highest in this case (highest observed value of the NIE metric at $0.062 \pm 0.001$), which is related to the highest representation harm observed between them (RH metric is $0.78 \pm 0.009$). Furthermore, the harm in CL representations causes a significant allocation harm for semantically similar pairs (`automobiles`, `trucks`), (`airplanes`, `ships`), and (`deers`, `horses`) as their NIE metrics are significantly higher. This aligns with our observations in Section 2.1, as these pairs suffer significant representation harm due to their underrepresentation.

Additionally, representation harm causes a significant reduction in downstream accuracy for these classes. This is observed in the diagonal NIE metrics in Figure 7, which is the highest for `deer` ($15.5 \pm 0.2\%$) and is at least $7.9 \pm 0.1\%$. This emphasizes that one cannot completely mitigate allocation harm without addressing the harm in CL representations.

For the BIASBIOS case study performing mediation analysis is challenging due to the existing natural imbalance, which makes it infeasible to create a balanced dataset and thus train a balanced CL model. It is, however, possible to quantify allocation harm, which we investigate in Appendix B.4.

## 5 SUMMARY AND DISCUSSION

We studied the effects of underrepresentations on contrastive learning algorithms empirically on the CIFAR10 and BIASBIOS datasets and theoretically in a stochastic block model. We find (both theoretically and empirically) that the CL representations of an underrepresented group collapse *to a semantically similar group*. Although prior work shows that classifiers trained on top of CL representations is more robust to underrepresentation than supervised learning (Liu et al., 2021c) in terms of downstream allocation harms, we show that the CL representations themselves suffer from representation harms: the CL representations of an underrepresented group collapse to a semantically similar group. To reconcile our results with prior work, we decompose the downstream allocation harms in classifiers trained on top of CL representations into a direct effect and an indirect effect mediated by the CL representations via a causal mediation analysis. Our results show that it is necessary to address the representation harms in CL representations in order to eliminate allocation harms in classifiers built on top of CL representations.

**Attempts to mitigate representation harms in CL:** Broadly speaking, the issue of underrepresentation is in conflict with one of the key advantages of CL or SSL: their ability to use large uncurated datasets from the Internet. Our results suggest that broad adoption of SSL can lead to representations harms to underpriviledged groups. This corroborates recent empirical results on algorithmic biases (Johnson, 2022; Naik & Nushi, 2023) in prompt-guided generative models powered by CLIP (Radford et al., 2021) (e.g., DALL·E 2), which uses a variant of CL loss. To address these issues, we need SSL methods that can account for underrepresentation *without requiring group annotations*. Prior works have proposed several such methods (Liu et al., 2021c; Zhou et al., 2022; Assran et al., 2022), but their evaluation metrics are limited to downstream accuracy, and thus their effectiveness in mitigating representation harms remains unexplored. In Appendix A.3 we investigate the representations learned with one of these methods, boosted contrastive learning (BCL) (Zhou et al., 2022), which turns out not to be sufficiently effective to completely mitigate the representation harm. Thus, the elimination of representation harms in CL remains a pressing open problem.

Our theoretical analysis suggests that practitioners should consider a combination of the following strategies to mitigate harm in representation; (1) reweighing the loss to counteract underrepresentation and (2) a "surgery" on connectivity, both of which require group annotations. Since they are not available in most CL applications, it would be interesting to attempt to combine the two using a proxy for group annotations. There have been many efforts to improve performance in minority groups without group annotations in the supervised learning setting (Hashimoto et al., 2018; Liu et al., 2021b; Zeng et al., 2022), but it remains to be seen whether these techniques can be transferred to SSL.

## ACKNOWLEDGMENTS

This paper is based upon work supported by the NSF under grants no. 2027737 and 2113373.

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

## A    SUPPLEMENTARY DETAILS FOR UNDER-REPRESENTATION STUDY WITH CIFAR10

### A.1    SUPPLEMENTARY DETAILS FOR SIMCLR

We use the implementation in `simclr.py` file of https://github.com/p3i0t/SimCLR-CIFAR10 for the training of contrastive learning (CL) models with the SimCLR training protocol. Please see `simclr.py` in supplementary codes for parameter values. We use the same parameters in both training cases with balanced and imbalanced datasets.

#### A.1.1    REPRESENTATION AND ALLOCATION HARM

The representation and allocation harms for all 10 classes are provided in Figures 8a and 8b. Here, each row corresponds to an underrepresentation to $1\%$ for the corresponding class. Additionally, in Figures 9a and 9b we present similar plots when classes are underrepresented at $5\%$ of their original sizes.

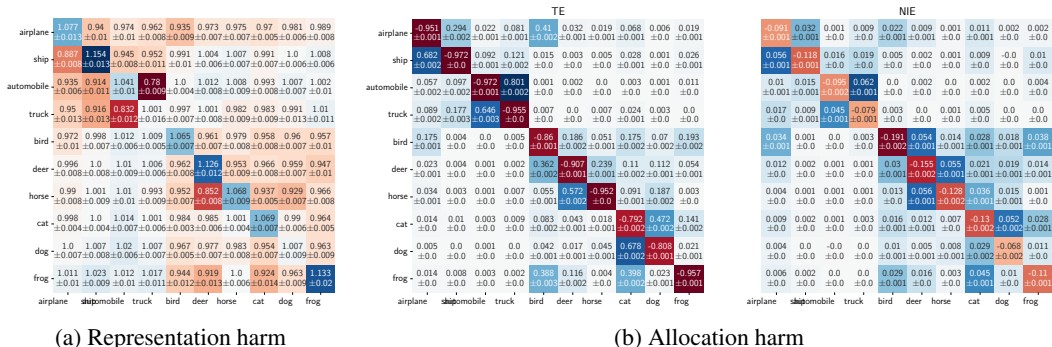

(a) Representation harm

(b) Allocation harm

Figure 8: Representation harm and total effect and natural indirect effect for allocation harm for SimCLR representations in CIFAR10 where the classes are undersampled to 1%.

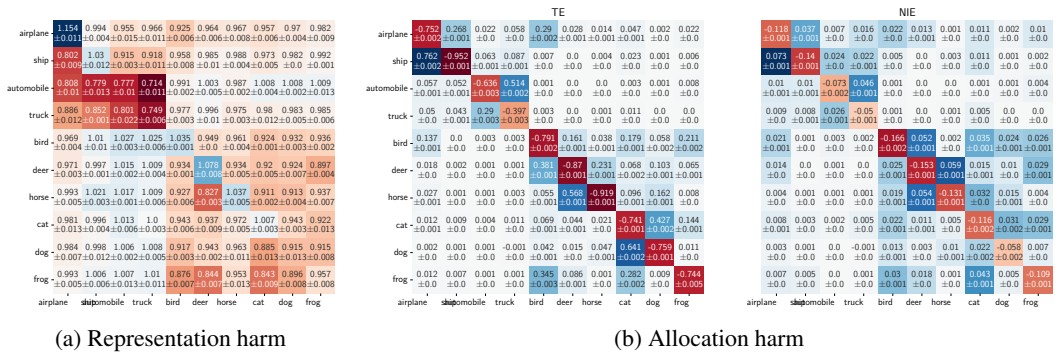

(a) Representation harm

(b) Allocation harm

Figure 9: Representation harm and total effect and natural indirect effect for allocation harm for SimCLR representations in CIFAR10 where the classes are undersampled to 5%.

## A.2 SUPPLEMENTARY DETAILS FOR SIMSIAM

We use the implementation in `main.py` file of https://github.com/Reza-Safdari/SimSiam-91.9-top1-acc-on-CIFAR10 for the training of CL models with the SimSIAM protocol. Please see our `jobs.py` and `main.py` for the specification of hyperparameters, which are kept the same in both training cases with balanced and imbalanced datasets. We use 3 repetitions of each setting with three different seed values. The rest of the details are the same as in SimCLR training.

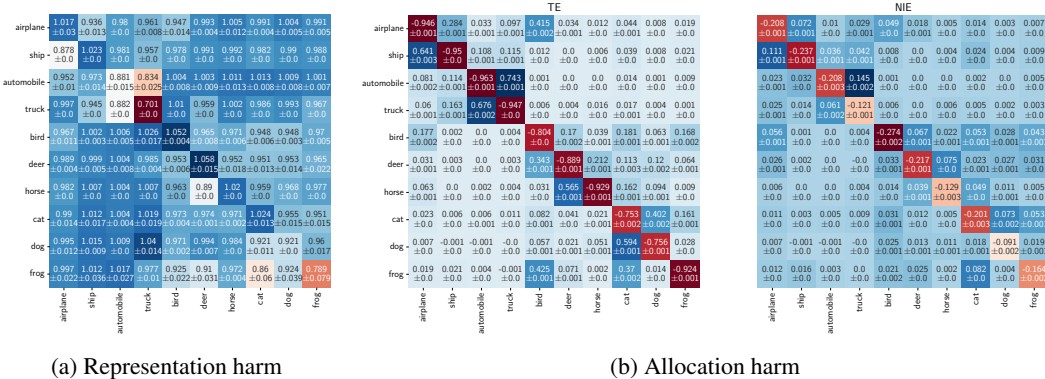

(a) Representation harm

(b) Allocation harm

Figure 10: Representation harm and total effect and natural indirect effect for allocation harm for SimSIAM representations in CIFAR10 over three repetitions.

### A.2.1 Representation and allocation harm

**Representation harm:** We plot the representation harm metrics equation 2.1 in Figure 10a. Similar to SimCLR representations in Section 2.1.2, representation harm is observed between semantically similar pairs (automobiles, trucks) (RH between them are $0.834 \pm 0.025$ and $0.882 \pm 0.0$), (airplanes, ships) (RH is $0.878 \pm 0.0$) and (deer, horse) (RH is $0.827 \pm 0.003$).

**Allocation harm:** In Figure 10b we plot the TE and NIE metrics equation 4.2, where we observe that representation harm causes allocation harm in a downstream classification. This is evident from the NIE metrics in Figure 10b, as the NIE between pairs (airplane, ship) are $0.111 \pm 0.001$ and $0.072 \pm 0.001$, between (automobile, truck) are $0.145 \pm 0.002$ and $0.061 \pm 0.002$, and finally between (deer, horse) are $0.075 \pm 0.0$ and $0.039 \pm 0.001$. Additionally, these classes suffer a non-trivial reduction in their prediction accuracies due to harm in representation. These observations are similar to our findings regarding allocation harm for SimCLR representation (Section 4).

### A.3 Boosted contrastive learning with SimCLR

We implement the boosted contrastive learning (BCL) algorithm (Zhou et al., 2022) on SimCLR protocol using the implementation in train.py file in https://github.com/MediaBrain-SJTU/BCL. Specifically, we use the BCL_I version of the algorithm, whose hyperparameter values can be found in the files jobs.py and train.py in the codes/cifar/BCL/ folder of our supplementary materials. Each setup is training for three repetitions with different seed values. The representation and allocation harm metrics for 10 CIFAR10 classes are provided in Figure 11.

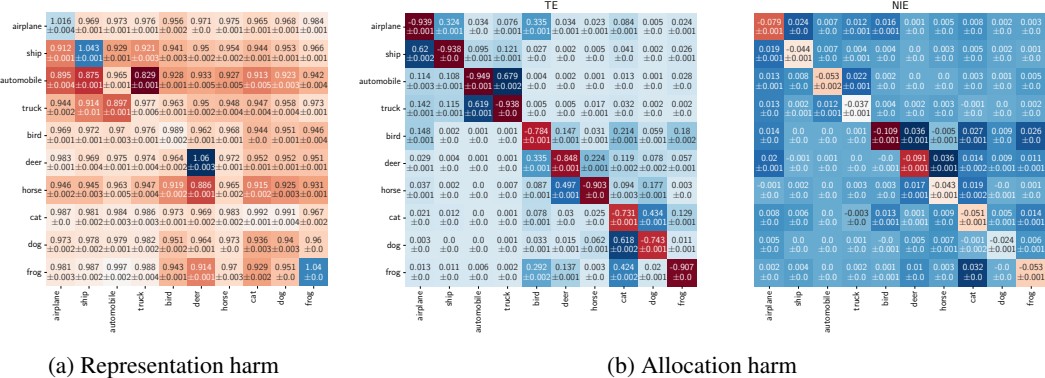

(a) Representation harm          (b) Allocation harm

Figure 11: Representation harm and total effect and natural indirect effect for allocation harm for BCL representations in CIFAR10 over three repetitions.

**Attempts to mitigate representation harms in CL with boosted CL (Zhou et al., 2022):** In our case study on CIFAR10 dataset in Figure 11, we find that BCL may not be sufficiently effective to completely mitigate the representation harm. Although the RH metrics between the pairs (automobiles, trucks) increase from $0.78 \pm 0.009$ and $0.832 \pm 0.012$ (in Figure 2) to $0.829 \pm 0.001$ and $0.897 \pm 0.001$ (in Figure 11a), BCL does not completely mitigate the harm in representations. As a result, accuracies for these classes still suffer non-trivial reductions, as observed in the diagonal NIE metrics in Figure 11b (the reduction in accuracy is $5.3 \pm 0.2\%$ (resp. $3.7 \pm 0.01\%$) for automobiles (resp. trucks)). These results show that the elimination of representation harms in CL remains a pressing open problem.

# B SUPPLEMENTARY DETAILS FOR UNDER-REPRESENTATION STUDY WITH BIASBIOS

## B.1 BIASBIOS DATASET DETAILS

In Figure 12, we provide the sample counts and frequencies for the 28 occupations and 2 genders (Male and Female) along with Total counts. We note that the dataset is imbalanced both in the occupation dimension as well as the gender dimension within a particular occupation. Particularly, the Professor occupation is the most occurring, while the Rapper occupation is the least occurring. Within each occupation, the proportion of samples of the two genders also varies mimicking societal gender stereotypes. For example, the occupations dietician, interior designer, model, nurse, etc. are dominated by Female samples, while the occupations composer, DJ, pastor, rapper, software engineer, surgeon, etc. are dominated by Male samples.

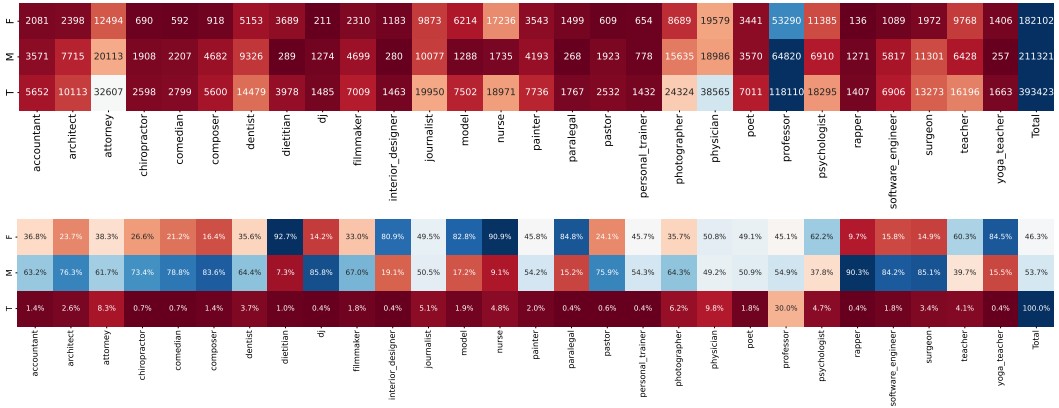

Figure 12: BIASBIOS dataset counts (above) and frequencies (below) by occupation and gender. The Y-axis labels are genders (M)ale, (F)emale, and (T)otal counts.

## B.2 SIMCSE EXPERIMENTAL SETUP

We randomly divide the ∼400k BIASBIOS dataset into the following three splits: 65% as training set, 10% as validation set, and 25% as test set. We use the official SimCSE implementation[1] to train the embedding model, with the optimal parameters used in the reported experiments listed in Table 1. To find these optimal parameters, we perform a grid search over the following parameters: training epochs $\in \{1, 3, 5\}$, learning rate $\in \{3e^{-5}, 1e^{-5}, 5e^{-5}\}$, batch size $\in \{64, 128, 256, 512\}$, and sequence lengths $\in \{32, 64, 128, 256, 512\}$, and then train a Logistic Regression model on top of each these embedding models. We select the parameters which result in the best occupation prediction accuracy on the validation set.

Table 1: Training parameters for SimCSE.

| model parameters | value |
|---|---|
| batch size | 64 |
| sequence length | 512 |
| learning rate | $1e^{-5}$ |
| training epochs | 1 |

## B.3 REPRESENTATION HARM

We present the GRH results for all occupation in the BIASBIOS dataset in Figure 13 (the F and M in the row labels indicate the under-represented gender for the corresponding occupation). We focus on the off-diagonal entries which should be >1 when there is no representation harm. We observe several deviations from this rule, broadly classifying into two types of deviations: (1) when representations of the under-represented group for an occupation collapse with a similar occupation, and (2) when representations of the under-represented group for an occupation collapses with several occupations. For the first type of deviation, we especially note GRH(architect (F), interior_designer) = 0.95, GRH(chiropractor (F), personal_trainer) = 0.96, GRH(nurse (M),

---

[1] https://github.com/princeton-nlp/SimCSE

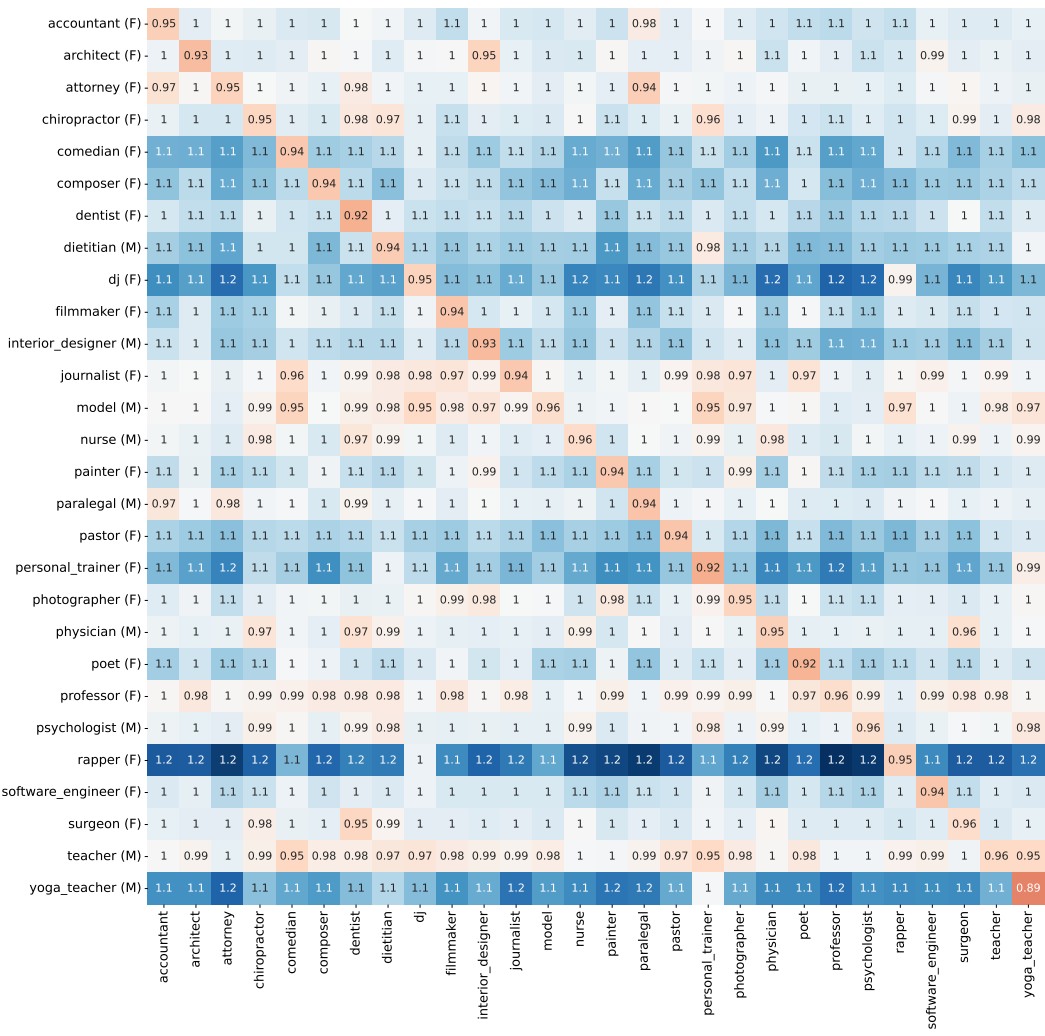

Figure 13: Gender RH for all occupations in BIASBIOS dataset.

`dentist) = 0.97`, `GRH(physician(M), surgeon) = 0.96`, and `GRH(surgeon (F), dentist) = 0.95` among others. These mentioned deviant groups are of occupations that are highly related. For the second type of deviation, we note that the `GRH` of Models (M), Journalists (F), and Teachers (M) have values $< 1$ for many occupations in the dataset.

## B.4 ALLOCATION HARM

We consider the task of occupation prediction from the SimCSE representations we learned on the BIASBIOS dataset using a logistic regression model trained on the same data as we used to learn the representations. Such a decision-making system may be used to assist in recruiting or hiring, applications where allocation harm can exacerbate gender disparity. To counteract the effect of under-representation at the supervised learning stage (thus focusing our attention on the effect of under-representation on CL), we reweigh the samples to achieve gender parity within each occupation when training the logistic regression following prior work (Idrissi et al., 2022; Sagawa et al., 2020). See Appendix B.5 for results with other weighting strategies.

**Metric:**  We define gender allocation harm (GAH) similarly to Equalized Odds, a popular group fairness criteria in the algorithmic fairness literature (Hardt et al., 2016):

$$\text{GAH}(l, m) = P(\hat{y} = m \mid y = l, \text{female}) - P(\hat{y} = m \mid y = l, \text{male}). \tag{B.1}$$

GAH can be understood as the difference of confusion matrices corresponding to each gender with entries far from 0 implying allocation harm. The average of the absolute values of the GAH diagonal is a common way to quantify violations of Equalized Odds.

**Results:**  In Figure 14 we present the GAH results for a subset of occupations. The complete set of GAH results is provided in Appendix B.5. Inspecting the diagonal entries, we see large performance gaps between genders for occupations such as `rapper`, `DJ`, and `model` (within each of these occupations the majority gender corresponds to at least 85% of the samples; these occupations are also underrepresented in the data with representation ranging from 0.4% for `rapper` to 1.9% for `model`). These results demonstrate that reweighing for gender parity when training the predictor may be insufficient to repair harms due to underrepresentation at the CL stage of the pipeline (it does, however, help to mitigate some of the biases of the unweighted model, as we show in the Appendix B.5). We also note the mistake patterns for related occupations: female

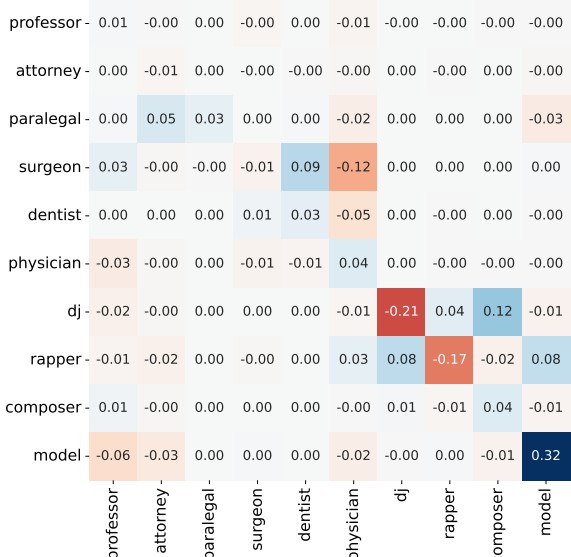

Figure 14: Gender AH in BIASBIOS dataset.

DJs are predicted as composers a lot more often than male ones, and female surgeons tend to be mistaken for dentists while male surgeons for physicians, despite similar performance on surgeons across genders. The `surgeon` example demonstrates that it may be insufficient to compare only class accuracies across genders (as is often done to measure group fairness violations via Equalized Odds) when quantifying allocations harms.

Compared to gender representation harms, we note that while the two have some overlap in terms of genders and occupations they are affecting, there are also differences, e.g., representations for female attorneys are closer to female paralegals than to male attorneys, but it does not manifest in the allocation harm analysis. We hypothesize that differences in which groups were affected by allocation and representation harms in our experiments might be due to the logistic regression model used for prediction utilizing only a subset (or subspace) of features most relevant to the task, while representation harm analysis takes into account all features.

The allocation harm does not have to be gender-specific. Similar to our CIFAR10 case study, samples from an underrepresented occupation can also be mistaken for a related occupation at a similar rate between genders. We observe this for the `paralegal` occupation which corresponds to about 0.4% of the training samples. Despite the gender imbalance (85% of paralegals in the data are female), both male and female class accuracy is the worst across occupations (14% for females and 11% for males) and the majority of them (66% for females and 61% for males) is predicted as attorneys, which is a more frequent class (8.3% of the samples are attorneys; see Appendix B for extended allocation harm analysis for occupations).

Overall, similar to the CIFAR10experiment, underrepresentation leads to the allocation harms of mistaking underrepresented groups for a related group. In some cases, the related group may be the same for both genders (e.g., in the case of paralegals), but in others, it can differ across genders (e.g., in the case of surgeons and DJs). Both cases can cause allocation harms for people from underrepresented occupations, whereas the latter additionally exacerbates gender stereotypes.

### B.5 OTHER DETAILS FOR ALLOCATION HARM

We experiment with three weighting strategies to counter the gender imbalance in the BIASBIOS dataset, namely: (1) Each sample is equally weighted, (2) We balance for gender imbalance within each class by weighting each sample as $W = \frac{N_y}{2N_y^g N}$ where $N$ is the total number of samples in the dataset, $N_y$ is the number of samples within a class and $N_y^g$ is the number of samples of gender $g$ in class $y$, and (3) We balance for gender and class imbalance within the dataset by weighting each sample as $W = \frac{1}{2*|y|*N_y^g}$ where $|y|$ is the number of classes and $N_y^g$ is the number of samples of gender $g$ in class $y$.

In Figure 15 we provide the Gender Allocation Harms for all occupations in the BIASBIOS dataset and for the three weighting strategies. Similar to our observations in Section 2.2, inspecting the diagonal entries, we see large performance gaps between genders of the same occupation across all three weighting strategies. We do, however, see that reweighing for gender parity (Figure 15b) does mitigate gender bias to a certain extent. For example, for the occupation `nurse` the GAH is mitigated, but it is merely reduced for `DJ` and `model`. Balancing for class frequencies in addition to gender (Figure 15c) has little effect on the GAH.

In Figure 16 we present confusion matrices for the three weighting strategies to quantify allocation harms (AH) for occupations (irrespective of gender). For occupations such as `paralegal` and `interior designer` we observed a small amount of GAH, however, we see that performance on these occupations is poor for the unweighted and gender-balanced weighting strategies (Figures 16a and 16b), as they are often confused with the related occupations (`attorney` and `architect` respectively). In this case, the representation harm is due to the under-representation of these occupations as opposed to the gender imbalance within occupations that we observed in the GAH experiments. Accounting for class imbalance in the weighting strategy helps to mitigate some of these biases (Figure 16c), e.g., the AH for `interior designer` is largely mitigated, while AH for `paralegal` is reduced.

Overall, we conclude that allocation harms due to under-representation in the contrastive learning stage can be *partially* mitigated during the supervised learning stage by curating/reweighing the data to have equal representation of classes and groups.

## C  A PROOF OF THEOREM 3.2

We recall the CL loss equation 3.2 and restate the theorem 3.2 for the reader's convenience. The minimization problem in a layer-peeled setting is

$$\min_{v_i \in \mathbf{S}^{d-1}} \mathbf{L}_{\mathrm{CL}}(V),$$

$$\mathbf{L}_{\mathrm{CL}}(V) \triangleq -\frac{1}{n} \sum_{i \in [n]} \frac{1}{\sum_{j \in [n]} e(i,j)} \sum_{j \in [n]} e(i,j) \log \left\{ \frac{\exp(v_i^\top v_j/\tau)}{\frac{1}{n} \sum_{l \in [n]} \exp(v_i^\top v_l/\tau)} \right\} \tag{C.1}$$

where $\mathbf{S}^{d-1}$ is the unit sphere in $\mathbb{R}^d$.

**Theorem C.1.** *At $n \to \infty$ the optimum representations obtained from the minimization of CL loss in equation 3.2 satisfy the following: $v_i^\star \overset{a.s.}{=} h_{Y_i}^\star$, where $\overset{a.s.}{=}$ denotes almost sure equality and $\{h_k^\star\}_{k \in [K]}$*

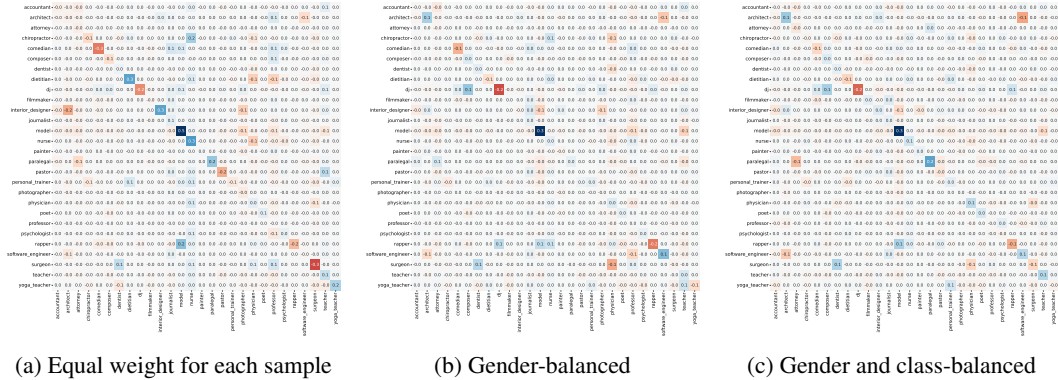

(a) Equal weight for each sample     (b) Gender-balanced     (c) Gender and class-balanced

Figure 15: Gender AH for all occupations in BIASBIOS across the three weighting strategies.

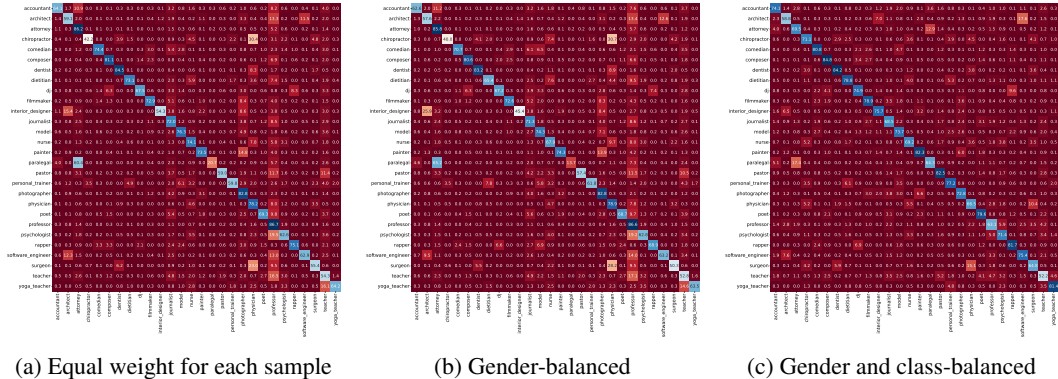

(a) Equal weight for each sample     (b) Gender-balanced     (c) Gender and class-balanced

Figure 16: AH for all occupations in BIASBIOS across the three weighting strategies.

*is a minimizer for*

$$\min_{h_k \in \mathbf{S}^{d-1}} -\sum_{k_1=1}^{K} \pi_{k_1} \frac{\sum_{k_2=1}^{K} \pi_{k_2} \alpha_{k_1,k_2} \frac{h_{k_1}^{\top} h_{k_2}}{\tau}}{\sum_{k_2=1}^{K} \pi_{k_2} \alpha_{k_1,k_2}} + \sum_{k_1=1}^{K} \pi_{k_1} \log \left\{ \sum_{k_3=1}^{K} \pi_{k_3} e^{h_{k_1}^{\top} h_{k_3}/\tau} \right\}. \quad \text{(C.2)}$$

*Note that the objective in equation 3.3 is a weighted version of the node2vec objective in equation 3.2 applied to common group-wise representations $h_{Y_i}$.*

For our convenience, we index the $i$-th node in $k$-th block as $(k, i)$ where $i \in [n_k]$ and $k \in [K]$. We start with a simplification of the CL loss.

### C.1 A SIMPLIFICATION OF THE LOSS

We divide the loss into two parts: a linear and a logarithmic sum exponential part.

$$\mathbf{L}_{\text{CL}}(V) \triangleq \frac{1}{n} \sum_{(k_1,i_1)} \frac{1}{d(k_1,i_1)} \sum_{(k_2,i_2)} -e_{(k_1,i_1),(k_2,i_2)} \log \left\{ \frac{\exp(\frac{v_{k_1,i_1}^{\top} v_{k_2,i_2}}{\tau})}{\frac{1}{n} \sum_{(k_3,i_3)} \exp(\frac{v_{k_1,i_1}^{\top} v_{k_3,i_3}}{\tau})} \right\}$$

$$= \underbrace{-\frac{1}{n} \sum_{(k_1,i_1)} \frac{1}{d(k_1,i_1)} \sum_{(k_2,i_2)} e_{(k_1,i_1),(k_2,i_2)} \left\{ \frac{v_{k_1,i_1}^{\top} v_{k_2,i_2}}{\tau} \right\}}_{\triangleq \mathbf{L}_{\text{linear}}(V)} \quad \text{(C.3)}$$

$$+ \underbrace{\frac{1}{n} \sum_{(k_1,i_1)} \frac{1}{d(k_1,i_1)} \sum_{(k_2,i_2)} \log \left\{ \frac{1}{n} \sum_{(k_3,i_3)} \exp(\frac{v_{k_1,i_1}^{\top} v_{k_3,i_3}}{\tau}) \right\}}_{\triangleq \mathbf{L}_{\text{LSE}}(V)}.$$

We simplify the log-sum exponential part as

$$
\begin{aligned}
\mathbf{L}_{\mathrm{LSE}}(V) &= \tfrac{1}{n}\textstyle\sum_{(k_1,i_1)}\tfrac{1}{d(k_1,i_1)}\sum_{(k_2,i_2)}\log\big\{\tfrac{1}{n}\sum_{(k_3,i_3)}\exp\big(\tfrac{v_{k_1,i_1}^\top v_{k_3,i_3}}{\tau}\big)\big\}\\
&= \tfrac{1}{n}\textstyle\sum_{(k_1,i_1)}\log\big\{\tfrac{1}{n}\sum_{(k_3,i_3)}\exp\big(\tfrac{v_{k_1,i_1}^\top v_{k_3,i_3}}{\tau}\big)\big\}\,.
\end{aligned}
\tag{C.4}
$$

For $k$-th group, we define $\hat{\pi}_k \triangleq \tfrac{n_k}{n}$ as the sample proportion and $h_k \triangleq \tfrac{1}{n_k}\sum_{i=1}^{n_k} v_{k,i}$ as the sample average of the representation vectors. We further define

$$
\widetilde{\mathbf{L}}_{\mathrm{linear}}(V) \triangleq \tfrac{1}{\tau}\sum_{k\in[K]}\hat{\pi}_k\frac{\sum_{k'\in[K]}\hat{\pi}_{k'}\alpha_{k,k'}h_k^\top h_{k'}}{\sum_{k'\in[K]}\hat{\pi}_{k'}\alpha_{k,k'}}
$$

and a modified loss:

$$
\check{\mathbf{L}}_{\mathrm{CL}}(V) \triangleq -\widetilde{\mathbf{L}}_{\mathrm{linear}}(V) + \mathbf{L}_{\mathrm{LSE}}(V)\,.
\tag{C.5}
$$

According to Lemma C.3 the linear part of the loss has the following $\ell_2$-convergence: as $n \to \infty$

$$
\max_V \big|\mathbf{L}_{\mathrm{linear}}(V) - \widetilde{\mathbf{L}}_{\mathrm{linear}}(V)\big| \overset{a.s.}{\to} 0\,,
\tag{C.6}
$$

and thus

$$
\begin{aligned}
&\max_V \big|\mathbf{L}_{\mathrm{CL}}(V) - \widetilde{\mathbf{L}}_{\mathrm{CL}}(V)\big|\\
&= \max_V \big|\mathbf{L}_{\mathrm{linear}}(V) + \mathbf{L}_{\mathrm{LSE}}(V) - \widetilde{\mathbf{L}}_{\mathrm{linear}}(V) - \mathbf{L}_{\mathrm{LSE}}(V)\big|\\
&= \max_V \big|\mathbf{L}_{\mathrm{linear}}(V) - \widetilde{\mathbf{L}}_{\mathrm{linear}}(V)\big| \overset{a.s.}{\to} 0\,.
\end{aligned}
\tag{C.7}
$$

In Sections C.2, C.3, and C.4 we perform a finite sample analysis on the modified CL loss in eq. equation C.5 and establish its neural collapse property.

## C.2  A LOWER BOUND

For studying the minimization of modified CL loss in equation C.5 we derive an achievable lower bound for $\check{\mathbf{L}}_{\mathrm{CL}}(V)$. We expand the log-sum exponential part

$$
\begin{aligned}
\mathbf{L}_{\mathrm{LSE}}(V) &= \tfrac{1}{n}\textstyle\sum_{(k_1,i_1)}\log\big\{\tfrac{1}{n}\sum_{(k_3,i_3)}\exp\big(\tfrac{v_{k_1,i_1}^\top v_{k_3,i_3}}{\tau}\big)\big\}\\
&= \tfrac{1}{n}\textstyle\sum_{(k_1,i_1)}\log\big\{\tfrac{1}{n}\sum_{(k_3,i_3)}\exp\big(\tfrac{h_{k_1}^\top h_{k_3}}{\tau}\big)\exp\big(\tfrac{v_{k_1,i_1}^\top v_{k_3,i_3}-h_{k_1}^\top h_{k_3}}{\tau}\big)\big\}\\
&= \tfrac{1}{n}\textstyle\sum_{(k_1,i_1)}\log\bigg\{\frac{\tfrac{1}{n}\sum_{(k_3,i_3)}\exp\big(\tfrac{h_{k_1}^\top h_{k_3}}{\tau}\big)\exp\big(\tfrac{v_{k_1,i_1}^\top v_{k_3,i_3}-h_{k_1}^\top h_{k_3}}{\tau}\big)}{\tfrac{1}{n}\sum_{(k_3,i_3)}\exp\big(\tfrac{h_{k_1}^\top h_{k_3}}{\tau}\big)}\bigg\}\\
&\quad + \tfrac{1}{n}\textstyle\sum_{(k_1,i_1)}\log\big\{\tfrac{1}{n}\sum_{(k_3,i_3)}\exp\big(\tfrac{h_{k_1}^\top h_{k_3}}{\tau}\big)\big\}\\
&= \tfrac{1}{n}\textstyle\sum_{(k_1,i_1)}\log\bigg\{\frac{\tfrac{1}{n}\sum_{(k_3,i_3)}\exp\big(\tfrac{h_{k_1}^\top h_{k_3}}{\tau}\big)\exp\big(\tfrac{v_{k_1,i_1}^\top v_{k_3,i_3}-h_{k_1}^\top h_{k_3}}{\tau}\big)}{\sum_{k_3}\hat{\pi}_{k_3}\exp\big(\tfrac{h_{k_1}^\top h_{k_3}}{\tau}\big)}\bigg\}\\
&\quad + \textstyle\sum_{k_1}\hat{\pi}_{k_1}\log\big\{\sum_{k_3\in[K]}\hat{\pi}_{k_3}\exp\big(\tfrac{h_{k_1}^\top h_{k_3}}{\tau}\big)\big\}
\end{aligned}
\tag{C.8}
$$

and provide a Jensen's inequality-based lower bound for

$$
\tfrac{1}{n}\textstyle\sum_{(k_1,i_1)}\log\bigg\{\frac{\tfrac{1}{n}\sum_{(k_3,i_3)}\exp\big(\tfrac{h_{k_1}^\top h_{k_3}}{\tau}\big)\exp\big(\tfrac{v_{k_1,i_1}^\top v_{k_3,i_3}-h_{k_1}^\top h_{k_3}}{\tau}\big)}{\sum_{k_3}\hat{\pi}_{k_3}\exp\big(\tfrac{h_{k_1}^\top h_{k_3}}{\tau}\big)}\bigg\}\,.
$$

Since $\log$ is a strictly concave function, we apply Jensen's inequality that $\log(\mathbb{E}[X]) \geq \mathbb{E}[\log X]$ to each summand of $(k_1,i_1)$ with respect to the probability measure

$$
\tfrac{1}{n}\textstyle\sum_{(k_3,i_3)}\frac{\exp(h_{k_1}^\top h_{k_3}/\tau)}{\sum_{k'}\hat{\pi}_{k'}\exp(h_{k_1}^\top h_{k'}/\tau)}\delta_{k_3,i_3}
$$

where $\delta_a$ is the Dirac-delta measure at $a$, (*i.e.* $X = \exp\left(\frac{h_{k_1}^\top h_{k_3}}{\tau}\right)$) and obtain

$$\log\left\{\frac{\frac{1}{n}\sum_{(k_3,i_3)} e^{h_{k_1}^\top h_{k_3}/\tau}\exp\left(\frac{v_{k_1,i_1}^\top v_{k_3,i_3} - h_{k_1}^\top h_{k_3}}{\tau}\right)}{\frac{1}{n}\sum_{(k_3,i_3)}\exp\left(\frac{h_{k_1}^\top h_{k_3}}{\tau}\right)}\right\} \tag{C.9}$$

$$\geq \frac{1}{n}\sum_{(k_3,i_3)}\left\{\frac{\exp(h_{k_1}^\top h_{k_3}/\tau)}{\sum_{k_3}\hat{\pi}_{k_3}\exp(h_{k_1}^\top h_{k_3}/\tau)}\right\}\left(\frac{v_{k_1,i_1}^\top v_{k_3,i_3} - h_{k_1}^\top h_{k_3}}{\tau}\right)$$

which we combine over $(k_1, i_1)$ to obtain the following

$$\frac{1}{n}\sum_{(k_1,i_1)}\log\left\{\frac{\frac{1}{n}\sum_{(k_3,i_3)} e^{h_{k_1}^\top h_{k_3}/\tau}\exp\left(\frac{v_{k_1,i_1}^\top v_{k_3,i_3} - h_{k_1}^\top h_{k_3}}{\tau}\right)}{\frac{1}{n}\sum_{(k_3,i_3)} e^{h_{k_1}^\top h_{k_3}/\tau}}\right\}$$

$$\geq \frac{1}{n}\sum_{k_1,i_1}\frac{1}{n}\sum_{k_3,i_3}\left\{\frac{\exp(h_{k_1}^\top h_{k_3}/\tau)}{\sum_{k_3}\hat{\pi}_{k_3}\exp(h_{k_1}^\top h_{k_3}/\tau)}\right\}\left(\frac{v_{k_1,i_1}^\top v_{k_3,i_3} - h_{k_1}^\top h_{k_3}}{\tau}\right) \tag{C.10}$$

$$= \sum_{k_1}\hat{\pi}_{k_1}\sum_{k_3}\hat{\pi}_{k_3}\left\{\frac{\exp(h_{k_1}^\top h_{k_3}/\tau)}{\sum_{k_3}\hat{\pi}_{k_3}\exp(h_{k_1}^\top h_{k_3}/\tau)}\right\}\left(\frac{\frac{1}{n_{k_1}}\sum_{i_1}\frac{1}{n_{k_3}}\sum_{i_3} v_{k_1,i_1}^\top v_{k_3,i_3} - h_{k_1}^\top h_{k_3}}{\tau}\right) = 0\,.$$

The above simplification implies

$$\mathbf{L}_{\mathrm{LSE}}(V) \geq \sum_{k_1}\hat{\pi}_{k_1}\log\left\{\sum_{k_3}\hat{\pi}_{k_3}\exp\left(\frac{h_{k_1}^\top h_{k_3}}{\tau}\right)\right\} \triangleq \widetilde{\mathbf{L}}_{\mathrm{LSE}}\,, \tag{C.11}$$

and thus

$$\check{\mathbf{L}}_{\mathrm{CL}}(V) = -\widetilde{\mathbf{L}}_{\mathrm{linear}}(V) + \mathbf{L}_{\mathrm{LSE}}(V) \geq -\widetilde{\mathbf{L}}_{\mathrm{linear}}(V) + \widetilde{\mathbf{L}}_{\mathrm{LSE}}(V) \triangleq \widetilde{\mathbf{L}}_{\mathrm{CL}}(V)\,. \tag{C.12}$$

### C.3  EQUALITY CONDITION OF LOWER BOUND

Under what condition equality is achieved in the inequality equation C.12? To answer that we look at the Jensen's inequality applied in equation C.9. Since $\log$ is a strictly concave function, equality is achieved when for any $(k_3, i_3)$ it holds

$$v_{k_1,i_1}^\top v_{k_3,i_3} - h_{k_1}^{\ \top} h_{k_3} = c_{k_1,i_1}\,, \tag{C.13}$$

where $c_{k_1,i_1} \in \mathbb{R}$ is the constant associated with the equality constraint for Jensen's inequality for $(k_1, i_1)$. Next, we establish that these constants $c_{k_1,i_1}$ must be exactly equal to zero. For this purpose, we average over both $i_1$ and $i_3$ in equation C.13

$$\frac{1}{n_{k_1}}\sum_{i_1}\frac{1}{n_{k_3}}\sum_{i_3} c_{k_1,i_1} = \frac{1}{n_{k_1}}\sum_{i_1=1}^{n_{k_1}}\frac{1}{n_{k_3}}\sum_{i_3=1}^{n_{k_3}}\{v_{k_1,i_1}^\top v_{k_3,i_3} - h_{k_1}^{\ \top} h_{k_3}\}$$

$$= h_{k_1}^{\ \top} h_{k_3} - h_{k_1}^{\ \top} h_{k_3} = 0\,. \tag{C.14}$$

A further simplification of the above equation leads to

$$\frac{1}{n_{k_1}}\sum_{i_1} c_{k_1,i_1} = 0\,. \tag{C.15}$$

Then, we let $k_1 = k_3 = k$ and $i_1 = i_3 = i$ in equation C.13 and average over $i$ to obtain

$$0 = \frac{1}{n_k}\sum_{i=1}^{n_k} c_{k,i}$$

$$= \frac{1}{n_k}\sum_i v_{k,i}^\top v_{k,i} - h_k^{\ \top} h_k \tag{C.16}$$

$$= \frac{1}{n_k}\sum_i (v_{k,i} - h_k)^\top (v_{k,i} - h_k) = \frac{1}{n_k}\sum_i \|v_{k,i} - h_k\|_2^2$$

which finally concludes that at the equality

$$v_{k,i} = h_k\,. \tag{C.17}$$

We further notice that $v_{k,i} \in \mathbf{S}^{d-1}$ which implies $h_k \in \mathbf{S}^{d-1}$.

### C.4  MINIMIZATION OF EQ. EQUATION C.5

Further developing on eq. equation C.12 we provide a final lower bound for eq. equation C.5.

$$\check{\mathbf{L}}_{\mathrm{CL}}(V) \geq \widetilde{\mathbf{L}}_{\mathrm{CL}}(V)$$

$$= -\frac{1}{\tau}\sum_{k\in[K]}\hat{\pi}_k\frac{\sum_{k'\in[K]}\hat{\pi}_{k'}\alpha_{k,k'} h_k^\top h_{k'}}{\sum_{k'\in[K]}\hat{\pi}_{k'}\alpha_{k,k'}} + \sum_{k_1}\hat{\pi}_{k_1}\log\left\{\sum_{k_3}\hat{\pi}_{k_3}\exp\left(\frac{h_{k_1}^\top h_{k_3}}{\tau}\right)\right\} \tag{C.18}$$

$$\geq -\frac{1}{\tau}\sum_{k\in[K]}\hat{\pi}_k\frac{\sum_{k'\in[K]}\hat{\pi}_{k'}\alpha_{k,k'}\hat{h}_k^\top \hat{h}_{k'}}{\sum_{k'\in[K]}\hat{\pi}_{k'}\alpha_{k,k'}} + \sum_{k_1}\hat{\pi}_{k_1}\log\left\{\sum_{k_3}\hat{\pi}_{k_3}\exp\left(\frac{\hat{h}_{k_1}^\top \hat{h}_{k_3}}{\tau}\right)\right\}$$

where

$$\{\hat{h}_k\} \in \arg\min_{h_k \in \mathbf{S}^{d-1}} \left\{ \begin{array}{l} -\frac{1}{\tau} \sum_{k \in [K]} \hat{\pi}_k \frac{\sum_{k' \in [K]} \hat{\pi}_{k'} \alpha_{k,k'} h_k^\top h_{k'}}{\sum_{k' \in [K]} \hat{\pi}_{k'} \alpha_{k,k'}} \\ + \sum_{k_1} \hat{\pi}_{k_1} \log \left\{ \sum_{k_3} \hat{\pi}_{k_3} \exp\left(\frac{h_{k_1}^\top h_{k_3}}{\tau}\right) \right\} \end{array} \right\}. \quad (C.19)$$

The lower bound in eq. equation C.18 does not involve any optimization variables ($v_{k,i}$ or $h_k$) and according to eq. equation C.17 the equality achieved when

$$v_{k,i} = \hat{h}_k. \quad (C.20)$$

Thus, at the minimum of $\check{\mathbf{L}}_{\mathrm{CL}}(V)$ it holds $v_{k,i} = \hat{h}_k$.

## C.5 FINAL CONCLUSION OF THEOREM 3.2

In Sections C.2, C.3, and C.4 we have established neural collapse for minimization of $\check{\mathbf{L}}_{\mathrm{CL}}(V)$. It remains to see how that translates to the minimization of $\mathbf{L}_{\mathrm{CL}}(V)$. In eq. equation C.7 we argued that

$$\max_V |\mathbf{L}_{\mathrm{CL}}(V) - \widetilde{\mathbf{L}}_{\mathrm{CL}}(V)| \overset{a.s.}{\to} 0. \quad (C.21)$$

Since both $\mathbf{L}_{\mathrm{CL}}(V)$ and $\check{\mathbf{L}}_{\mathrm{CL}}(V)$ are continuous, we use a continuous mapping theorem to conclude that

$$d\big(\arg\min_V \mathbf{L}_{\mathrm{CL}}(V), \arg\min_V \check{\mathbf{L}}_{\mathrm{CL}}(V)\big) \overset{a.s.}{\to} 0 \quad (C.22)$$

in any metric $d$ for measuring set difference, or, alternatively speaking

$$\arg\min_V \mathbf{L}_{\mathrm{CL}}(V) \overset{a.s.}{\to} \arg\min_V \check{\mathbf{L}}_{\mathrm{CL}}(V). \quad (C.23)$$

Since, the loss in eq. equation C.19 convergences almost surely to the loss in eq. equation 3.3 we conclude the Theorem 3.2.

## C.6 ADDITIONAL LEMMA

**Lemma C.2.** *Consider a generic sequence of vectors $\{\xi_{k,i}\}_{k \in [K], i \in [n_k]} \subset \mathbf{S}^{p-1}$ and define the group mean $\bar{\xi}_k = \frac{1}{n_k} \sum_i \xi_{k,i}$ and*

$$\epsilon(\Xi) \triangleq \frac{1}{n} \sum_{(k_2,i_2)} e_{(k_1,i_1),(k_2,i_2)} \xi_{k_2,i_2} - \sum_{k_2} \hat{\pi}_{k_2} \alpha_{k_1,k_2} \bar{\xi}_{k_2}. \quad (C.24)$$

*Then*

$$\max_\Xi \|\epsilon(\Xi)\|_2 \overset{a.s.}{\to} 0. \quad (C.25)$$

*Proof of the lemma C.2.* Indexing the dependence of $\epsilon(\Xi)$ as $\epsilon_n(\Xi)$ we notice that

$$\begin{aligned} \epsilon_n(\Xi) &= \frac{1}{n} \sum_{(k_2,i_2)} e_{(k_1,i_1),(k_2,i_2)} \xi_{k_2,i_2} - \sum_{k_2} \hat{\pi}_{k_2} \alpha_{k_1,k_2} \bar{\xi}_{k_2} \\ &= \frac{1}{n} \sum_{(k_2,i_2)} \big\{ e_{(k_1,i_1),(k_2,i_2)} - \alpha_{k_1,k_2} \big\} \xi_{k_2,i_2} \end{aligned} \quad (C.26)$$

and

$$\begin{aligned} \mathbb{E}[\epsilon_n(\Xi)^4] &= \mathbb{E}\big[\big\{ \frac{1}{n} \sum_{(k_2,i_2)} \big\{ e_{(k_1,i_1),(k_2,i_2)} - \alpha_{k_1,k_2} \big\} \xi_{k_2,i_2} \big\}^4 \big] \\ &= \frac{1}{n^4} \sum_{(k_2,i_2)} \mathbb{E}[\{ e_{(k_1,i_1),(k_2,i_2)} - \alpha_{k_1,k_2} \}^4] \\ &\quad + \frac{1}{n^4} \sum_{(k_2,i_2) \neq (k_3,i_3)} \mathbb{E}[\{ e_{(k_1,i_1),(k_2,i_2)} - \alpha_{k_1,k_2} \}^2] \mathbb{E}[\{ e_{(k_1,i_1),(k_3,i_3)} - \alpha_{k_1,k_3} \}^2] \\ &= O(\frac{1}{n^3}) + O(\frac{1}{n^2}) = O(\frac{1}{n^2}). \end{aligned} \quad (C.27)$$

Thus, for any $\delta > 0$ we have

$$\delta^4 \sum_{n \geq 1} P\big(\|\epsilon_n(\Xi)\|_2 > \delta\big) \leq \sum_{n \geq 1} \mathbb{E}[\epsilon_n(\Xi)^4] < \infty. \quad (C.28)$$

Next, we use the first Borel-Cantelli lemma to conclude that for any $\Xi$ we have

$$\|\epsilon_n(\Xi)\|_2 \overset{a.s.}{\to} 0. \quad (C.29)$$

Finally, we use the continuity of $\epsilon_n(\Xi)$ and separability of $\mathbf{S}^{p-1}$ to conclude the statement of the lemma.

$\square$

**Lemma C.3.** *Assume 3.1. Then as $n \to \infty$ the following convergence holds.*

$$\max_V \left| \mathbf{L}_{\text{linear}}(V) - \widetilde{\mathbf{L}}_{\text{linear}}(V) \right| \overset{a.s.}{\to} 0 \,. \tag{C.30}$$

*Proof of the Lemma C.3.* We start with an expansion of $\mathbf{L}_{\text{linear}}(V) - \widetilde{\mathbf{L}}_{\text{linear}}(V)$.

$$
\begin{aligned}
&\mathbf{L}_{\text{linear}}(V) - \widetilde{\mathbf{L}}_{\text{linear}}(V) \\
&= \tfrac{1}{n} \sum_{(k_1,i_1)} \tfrac{1}{d(k_1,i_1)} \sum_{(k_2,i_2)} e_{(k_1,i_1),(k_2,i_2)} \left\{ \tfrac{v_{k_1,i_1}^\top v_{k_2,i_2}}{\tau} \right\} \\
&\quad - \sum_{k_1} \hat{\pi}_{k_1} \tfrac{\sum_{k_2} \hat{\pi}_{k_2} \alpha_{k,k_2}}{\sum_{k' \in [K]} \hat{\pi}_{k'} \alpha_{k_1,k'}} \left\{ \tfrac{h_{k_1}^\top h_{k_2}}{\tau} \right\} \\
&= \tfrac{1}{n} \sum_{(k_1,i_1)} \tfrac{1}{d(k_1,i_1)} \sum_{(k_2,i_2)} e_{(k_1,i_1),(k_2,i_2)} \left\{ \tfrac{v_{k_1,i_1}^\top v_{k_2,i_2}}{\tau} \right\} \\
&\quad - \tfrac{1}{n} \sum_{(k_1,i_1)} \tfrac{\sum_{k_2} \hat{\pi}_{k_2} \alpha_{k,k_2}}{\sum_{k' \in [K]} \hat{\pi}_{k'} \alpha_{k_1,k'}} \left\{ \tfrac{v_{(k_1,i_1)}^\top h_{k_2}}{\tau} \right\} \\
&= \tfrac{1}{n\tau} \sum_{(k_1,i_1)} v_{(k_1,i_1)}^\top \left\{ \tfrac{\sum_{(k_2,i_2)} e_{(k_1,i_1),(k_2,i_2)} v_{k_2,i_2}}{d(k_1,i_1)} - \tfrac{\sum_{k_2} \hat{\pi}_{k_2} \alpha_{k_1,k_2} h_{k_2}}{\sum_{k' \in [K]} \hat{\pi}_{k'} \alpha_{k,k'}} \right\} \,.
\end{aligned} \tag{C.31}
$$

Following the expansion we define

$$
\begin{aligned}
\epsilon_1(V) &= \tfrac{1}{n} \sum_{(k_2,i_2)} e_{(k_1,i_1),(k_2,i_2)} v_{k_2,i_2} - \sum_{k_2} \hat{\pi}_{k_2} \alpha_{k_1,k_2} h_{k_2} \\
\epsilon_2 &= \tfrac{d(k_1,i_1)}{n} - \sum_{k'} \hat{\pi}_{k'} \alpha_{k_1,k'},
\end{aligned} \tag{C.32}
$$

and note that $\epsilon(V) = \epsilon_1(V)$ and $\epsilon(1) = \epsilon_2$, where $\epsilon(\cdot)$ is defined in lemma C.2. As a conclusion of the lemma we have

$$\max_V \|\epsilon_1(V)\|_2 \overset{a.s.}{\to} 0, \text{ and } |\epsilon_2| \overset{a.s.}{\to} 0 \,. \tag{C.33}$$

Next, we notice that

$$
\begin{aligned}
&\max_V \left| \mathbf{L}_{\text{linear}}(V) - \widetilde{\mathbf{L}}_{\text{linear}}(V) \right| \\
&\leq \max_V \tfrac{1}{\tau} \sum_{(k_1,i_1)} \hat{\pi}_{k_1} \|h_{k_1}\|_2 \cdot \left\| \tfrac{\sum_{(k_2,i_2)} e_{(k_1,i_1),(k_2,i_2)} v_{k_2,i_2}}{d(k_1,i_1)} - \tfrac{\sum_{k_2} \hat{\pi}_{k_2} \alpha_{k,k_2} h_{k_2}}{\sum_{k' \in [K]} \hat{\pi}_{k'} \alpha_{k,k'}} \right\|_2 \\
&\leq \tfrac{1}{\tau} \sum_{(k_1,i_1)} \hat{\pi}_{k_1} \cdot \max_V \left\| \tfrac{\tfrac{1}{n}\sum_{(k_2,i_2)} e_{(k_1,i_1),(k_2,i_2)} v_{k_2,i_2}}{\tfrac{d(k_1,i_1)}{n}} - \tfrac{\sum_{k_2} \hat{\pi}_{k_2} \alpha_{k,k_2} h_{k_2}}{\sum_{k' \in [K]} \hat{\pi}_{k'} \alpha_{k,k'}} \right\|_2,
\end{aligned} \tag{C.34}
$$

where we focus on each term within the sum

$$
\begin{aligned}
&\max_V \left\| \tfrac{\tfrac{1}{n}\sum_{(k_2,i_2)} e_{(k_1,i_1),(k_2,i_2)} v_{k_2,i_2}}{\tfrac{d(k_1,i_1)}{n}} - \tfrac{\sum_{k_2} \hat{\pi}_{k_2} \alpha_{k_1,k_2} h_{k_2}}{\sum_{k' \in [K]} \hat{\pi}_{k'} \alpha_{k_1,k'}} \right\|_2 \\
&= \max_V \left\| \tfrac{\sum_{k_2} \hat{\pi}_{k_2} \alpha_{k_1,k_2} h_{k_2} + \epsilon_1(V)}{\sum_{k' \in [K]} \hat{\pi}_{k'} \alpha_{k_1,k'} + \epsilon_2} - \tfrac{\sum_{k_2} \hat{\pi}_{k_2} \alpha_{k_1,k_2} h_{k_2}}{\sum_{k' \in [K]} \hat{\pi}_{k'} \alpha_{k_1,k'}} \right\|_2 \\
&= \max_V \tfrac{\left\| \epsilon_2 \{ \sum_{k_2} \hat{\pi}_{k_2} \alpha_{k_1,k_2} h_{k_2} \} + \epsilon_1(V) \{ \sum_{k' \in [K]} \hat{\pi}_{k'} \alpha_{k_1,k'} \} \right\|_2}{\left| \{ \sum_{k' \in [K]} \hat{\pi}_{k'} \alpha_{k_1,k'} + \epsilon_2 \} \cdot \{ \sum_{k' \in [K]} \hat{\pi}_{k'} \alpha_{k_1,k'} \} \right|} \\
&\leq \tfrac{|\epsilon_2| \cdot \max_V \left\| \sum_{k_2} \hat{\pi}_{k_2} \alpha_{k_1,k_2} h_{k_2} \right\|_2 + \{ \max_V \|\epsilon_1(V)\|_2 \} \cdot \{ \sum_{k' \in [K]} \hat{\pi}_{k'} \alpha_{k_1,k'} \}}{\left| \sum_{k' \in [K]} \hat{\pi}_{k'} \alpha_{k_1,k'} + \epsilon_2 \right| \cdot \left| \sum_{k' \in [K]} \hat{\pi}_{k'} \alpha_{k_1,k'} \right|} \,.
\end{aligned} \tag{C.35}
$$

Here, both $|\epsilon_2|$ and $\max_V \|\epsilon_1(V)\|_2$ convergence almost surely to zero. Since $\sum_{k' \in [K]} \pi_{k'} \alpha_{k_1,k'} > 0$ both $\sum_{k' \in [K]} \hat{\pi}_{k'} \alpha_{k_1,k'} > 0$ and $\sum_{k' \in [K]} \hat{\pi}_{k'} \alpha_{k_1,k'} + \epsilon_2 > 0$ for sufficiently large $n$. Finally,

$$\max_V \left\| \sum_{k_2} \hat{\pi}_{k_2} \alpha_{k_1,k_2} h_{k_2} \right\|_2 \leq \sum_{k_2} \hat{\pi}_{k_2} \alpha_{k_1,k_2} \{ \max_V \|h_{k_2}\|_2 \} \leq \sum_{k_2} \hat{\pi}_{k_2} \alpha_{k_1,k_2} \,. \tag{C.36}$$

Thus we have

$$\max_V \left\| \tfrac{\tfrac{1}{n}\sum_{(k_2,i_2)} e_{(k_1,i_1),(k_2,i_2)} v_{k_2,i_2}}{\tfrac{d(k_1,i_1)}{n}} - \tfrac{\sum_{k_2} \hat{\pi}_{k_2} \alpha_{k_1,k_2} h_{k_2}}{\sum_{k' \in [K]} \hat{\pi}_{k'} \alpha_{k_1,k'}} \right\|_2 \overset{a.s.}{\to} 0 \tag{C.37}$$

which proves the lemma that

$$\max_V \left| \mathbf{L}_{\text{linear}}(V) - \widetilde{\mathbf{L}}_{\text{linear}}(V) \right| \overset{a.s.}{\to} 0 \tag{C.38}$$

$\square$

## D    SUPPLEMENT FOR SBM SIMULATION IN SECTION 3.1

We generate an SBM dataset using `graspologic.simulations.sbm` function. We obtain the representation by optimizing the node2vec loss in eq. equation 3.1 using a gradient descend algorithm that has step size for $t$-th step as $\text{lr}_t = 0.1 \times t^{-0.2}$ and $T = 30000$ as the number of iterations.

