# OpenReview forum: "An Investigation of Representation and Allocation Harms in Contrastive Learning"
_ICLR.cc/2024/Conference — ICLR 2024 poster_

### Official Review · Reviewer_F3P4 · 2023-10-28

**Soundness:** 3 good
**Presentation:** 4 excellent
**Contribution:** 3 good
**Rating:** 8
**Confidence:** 3

**Summary:**

This paper focuses on studying the issue of representation harm in contrastive learning (CL) that arises when some groups are underrepresented in the training corpora. In this case, the authors show that the underrepresented groups tend to collapse into semantically similar groups (that are not underrepresented). In a follow up theoretical analysis on graphs, the authors show that two groups of nodes tend to collapse as their connectivity increases. Finally, through a causal analysis, the authors show that the representation harms caused by CL cannot be mitigated for downstream tasks when training a probe on top of the CL representations.

**Strengths:**

1. The paper studies an important problem in self-supervised learning through contrastive objectives (that of representation collapse of groups that are underrepresented in the training data)
2. It does so in a sound and systematic way, by providing evidence on controlled images (CIFAR10), imbalanced text (BiasBios) and a theoretical analysis with artificial graphs.
3. The paper is well written, with a thorough discussion of results and potential insufficiencies of an existing class of algorithms in overcoming representation harm, showing how more work is needed in the area of CL to result in algorithms that are robust to underrepresentation of certain groups (which can be hard to identify at scale to begin with).

**Weaknesses:**

The main weakness I can find from this paper is the lack of a large-scale study. SSL, and CL in particular, are most effective when used on large amounts of data. The controlled studies in this paper allow us to analyze the existing representation harms. However, it can be seen that these values are generally much lower for BiasBios than for CIFAR10. On the other hand, this could be due to the different domain (text). A study on ImageNet (larger number of both samples and classes) could potentially disentangle this confound. In particular, it would be interesting to know whether having more diversity in the data alleviates learning spurious features (e.g. collapsing classes with similar background colors), and reduces collapse of underrepresented groups.

**Questions:**

1. Typo “a” at the end of line 6 in page 2
2. You could use diverging palettes (with a neutral color at 1.0) in the heatmaps, to clearly distinguish HR<1. Having a colorbar next to each heatmap figure would also improve readability.
3. In Figure 2, do you have any idea why there’s such a large difference (10%) between deer and horse collapsing?
4. In line 4 of Sec 2.1.2, you can add “of classes” after “a pair” to help the reader understand better
5. In Sec 2.2.2, when you define the GRH metric, I would have found it useful to have a sketch of a plane with boundaries that show what each region means. It’s something you could consider adding when you get an extra page
6. In Sec 4.2, why do you train on 75% of the Test set?

---

> ### Author Response · Authors · 2023-11-16
> **Response to Reviewer F3P4**
>
> Thank you for your valuable feedback and questions. Please see our responses for the other comments.
>
> **However, it can be seen that these values are generally much lower for BiasBios than for CIFAR10.** We would like to emphasize that the results from the CIFAR10 and BiasBios datasets are not directly comparable, as they are based on slightly different metrics. The metric for the CIFAR10 dataset is specific to our *controlled study* design, where we measure the degree to which the representations of two classes have collapsed with each other due to under-representation by comparing representations learned using balanced data to those learned using imbalanced data. In our experiments with the BiasBios dataset, we wish to demonstrate the potential harmfulness of collapse in a real data scenario where *underrepresentation occurs naturally*, which requires a slightly different measure of collapse. Specifically, we measure the ratio of the similarity of samples with the same gender and different occupations to the similarity of samples with different gender and the same occupation. This measure allows us to notice harmful cases of collapse, such as female surgeons being more similar to female dentists than to male surgeons.
>
> **Regarding ImageNet:** Thank you for the suggestion. We started looking into such an experiment, but realized that the `imagenet-1k` dataset used for training publicly available CL models is indeed pretty balanced. Most classes have 1300 samples, with the least frequent class being `black-and-tan_coonhound` with 732 samples. Therefore, it is not suitable for studying the effect of underrepresentation. Artificially subsampling the data and training our own CL model as we did on CIFAR10 is unfortunately computationally too challenging for our computational resources.
>
> **Large difference between deer and horse collapse:** Although we are not completely certain, a possible explanation is that the representations of `deer` are more dispersed compared to `horse` when undersampled, as seen in Figure 2 that the diagonal RH for `deer` ($1.126 \pm 0.012$) is significantly higher than `horse` ($1.068 \pm 0.009$). Since `deer` becomes more scattered, the representation harm metric between `deer` and `horse` is less compared to when `horse` is undersampled.
>
> **Training on 75\% of the test set:** The linear heads are trained with a randomly chosen 75\% of the test dataset and evaluated with the *remaining* 25\% to calculate the metrics in eq. (4.2).
>
> **Typos and figures** Thank you for your suggestions. We have corrected the typos, are updating the figures, and will update the manuscript soon. Current heatmaps already use a divergent heatmap (1 is light green, above one are cold colors like blue, and below 1 are warm colors like red), but are not "centered". We will center the 1 to be white to improve readability.

---

> > ### Comment · Reviewer_F3P4 · 2023-11-23
> >
> > Thank you for the reply!
> >
> > **On ImageNet:** I think that having results on a sub-sampled version of ImageNet by the (potential) CR version would make the paper stronger.
> >
> > **Training on 75% of the test set:** I meant to ask, why don't you use a different split to begin with? And then test on 100% of the test set?

---

> > > ### Author Response · Authors · 2023-11-28
> > > **Response to reviewer F3P4**
> > >
> > > Thank you for your response and additional comments. Please see our responses below.
> > >
> > > **Training on 75% of the test set:** As you have suggested, technically one could train the linear head on the training dataset and test it on the full test dataset. However, in this case, the trained linear head and CL models become dependent with each other, because they are trained with the same training dataset. As we want to tease out the allocation harm in the biased CL model in a downstream classification task, we want to keep the linear head and the CL model independent of each other. Since CIAFR10 only has training and test splits and doesn't have a validation split, we train the CL model on training split, and use 75% of the test dataset for training the linear head.
> > >
> > > **On ImageNet:** As an alternative to ImageNet, we are considering an experiment with the CIFAR100 dataset, which has 100 classes, while being easier to experiment with in terms of data size given our computational resources. This dataset has reasonably large classes, allowing more diversity in the dataset than CIFAR10, similarly to what you suggested in the review. We intend to subsample one class from each of the superclasses and investigate whether the effects representation and allocation harms are as severe as in CIFAR10.

---

### Official Review · Reviewer_n65V · 2023-11-01

**Soundness:** 2 fair
**Presentation:** 3 good
**Contribution:** 3 good
**Rating:** 6
**Confidence:** 3

**Summary:**

This paper studies the effect of under-representation on the performance of minority groups in the context of self-supervised learning (SSL), specifically contrastive learning (CL). They show that CL tends to collapse representations of minority groups with certain majority groups, leading to representation harms and downstream allocation harms even when labeled data is balanced. Theory and experiments are presented to support their results.

**Strengths:**

1. This is a well written paper that discusses an important topic that is well motivated. This rigorous analysis is likely of interest to practitioners and useful to mitigate the potential harm from CL methods.
2. The empirical study is good and the theoretical study adds a solid foundation to the empirical results. Both empirical and theoretical findings are promising.
3. Section 4 is also quite interesting, showing how representation harms can cause allocation harms.

**Weaknesses:**

1. There needs to be an intuitive definition of allocative harms and representation harms when they are first mentioned in the intro, which matches the precise definition in section 2.
2. Figure text is generally too small to see clearly.
3. Why does this metric for representation harm make sense? It seems like an important decision critical for the rest of the paper, so needs better justification.
4. The examples used in section 2 are not very useful - sure automobiles and trucks could collapse but is it the worst thing - are there more compelling real-world examples to illustrate these problems?
5. Why are the metrics for representation harm in 2.1.2 and 2.2.2 different? This seems weird. Even if the exact distance (eg cosine vs l2 for image or word embeddings) are different - why are the metrics different?
6. Why do sections 5 and 6 require a different data setup? Specifically, why is causal mediation analysis the right framework to study section 6 - this was not motivated.
7. The empirical analysis is conducted only on CIFAR10 and BIASBIOS datasets - I would have liked to see more to further strengthen the results in the paper, such as including celebA which is quite standard for studying biases in vision models, or perhaps even with results on a CLIP model for results on image-text models.

**Questions:**

see weaknesses above

---

> ### Author Response · Authors · 2023-11-16
> **Response to Reviewer n65V**
>
> Thank you for your insightful comments and questions. Please see our responses below.
>
>
> **Intuitive definition of allocative harms and representation harms:**
> We added an intuitive definition and an example of allocative harms in the first paragraph of the Introduction. *Allocation harm* refers to the demographic disparities in resource allocations, and this harm is easily traced back to the rate of misclassification of the underlying resource allocation mechanism. In our eq. (4.2) we use this difference miclassification rate to quantify the allocation harm caused by under-representation.
>
>
> We also extended the second paragraph of the Introduction with an example of *representation harm* to facilitate the intuition behind it. In the third paragraph and in Figure 1 we also present an example using the CIFAR10 dataset, where an undersampling of the `automobile` images causes their representations to cluster with those of `trucks`. In Section 2.1.2, we measure this clustering by calculating the ratio of the average cosine distances between the representations of two groups when one of them is under-represented to the average distance between the same two groups when they are class-balanced.
>
>
> **Different representation harm metrics in Sections 2.1.2 and 2.2.2:** Although the exact quantification of representation harm metric may depend on specific instances of under-representation, the main idea behind the metrics are similar: they measure the collapse of the representations of an under-represented group with semantically similar groups. In our experiments with the CIFAR10 dataset in Section 2.1 we *control the undersampling* for each class and wish to demonstrate that the collapse grows/emerges when the underrepresentation occurs. In our experiments with the BiasBios dataset, we wish to demonstrate the potential harmfulness of collapse in a real data scenario where *underrepresentation occurs naturally*, which requires a slightly different measure of collapse. Specifically, we measure the ratio of the similarity of samples with the same gender and different occupations to the similarity of samples with different gender and the same occupation. This measure allows us to notice harmful cases of collapse, such as female surgeons being more similar to female dentists than to male surgeons.
>
>
> **Justification of the representation harm metric:** We reiterate that the key idea behind measuring representation harm is to measure *collapse* between groups in the data, while relevant measures of collapse, the definitions of groups, and the kinds of groups between which collapse is most problematic may vary across domains and applications.
>
>
> **Examples used in Section 2 are not very useful:** The collapse between `automobile` and `truck` representations (in the third paragraph of Section 1 and Section 2.1) is an example to illustrate the idea of representation harm. For a more realistic example, we refer to the Wall Street Journal article titled "Google Mistakenly Tags Black People as 'Gorillas,' Showing Limits of Algorithms" [3]. We have included this example in the second paragraph in the Introduction and in the first paragraph of Section 2.1.2. Additionally, we provide realistic examples in our experiments with the BiasBios dataset in Section 2.2.2, where the representations of under-represented female surgeons are closer to the representation of female dentists than they are to male surgeons. Similar harm is also observed between the `attorney` and `paralegal` professions. For a more detailed discussion, see our last paragraph in Section 2.2.2 named **Results**.
>
> **Why do sections 5 and 6 require a different data setup?** We assume that the reviewer meant Sections 3 and 4. If so, then we point out that the synthetic experiments in Section 3 are used to motivate and interpret our theoretical analysis (Section 3.2) and to draw its connections with our empirical studies in Section 2. On the other hand, the causal mediation analysis in Section 4 shows that in CIFAR10 the downstream allocation harm in the classification task is partly caused by the representation harm.
>
> **Motivation for causal framework:** As we explain in the second line of the first paragraph of Section 4 and in Section 4.1, causal mediation analysis [1] is the statistically principled way to dissect total allocation harm (**TE**) and tease out the part caused by representation harm through the natural indirect effect (**NIE**).

---

> > ### Author Response · Authors · 2023-11-16
> > **Response to Reviewer n65V continued**
> >
> > **Empirical experiments beyond the CIFAR10 and BIASBIOS datasets:** We believe that the experiments conducted with the two datasets, the synthetic dataset, and the theoretical analysis were sufficient to demonstrate that under-representation can lead to representation and allocation harm in contrastive learning settings. In addition to various CL methods, such as SimCLR, SimSiam, and SimCSE, we have also included the *boosted contrastive learning* method [2] (in Section 5) to further strengthen our experimental results. Although we acknowledge that additional experiments would be beneficial, the current paper already consists of nine pages of main text and 11 pages of supplementary material, not including references.
> >
> >
> >
> > **Enlarge texts in figures:** Thank you for the suggestion. We are updating the figures with larger text and shall update our draft with modified figures shortly.
> >
> >
> > ---
> > # References
> >
> > [1] Pearl, J. (2022). Direct and indirect effects. In Probabilistic and causal inference: the works of Judea Pearl (pp. 373-392).
> >
> > [2] Zhou, Z., Yao, J., Wang, Y. F., Han, B., & Zhang, Y. (2022, June). Contrastive learning with boosted memorization. In International Conference on Machine Learning (pp. 27367-27377). PMLR.
> >
> > [3] Barr, A. (2015). Google mistakenly tags black people as ‘gorillas,’showing limits of algorithms. The Wall Street Journal, 1(7), 2015.

---

> > > ### Author Response · Authors · 2023-11-20
> > >
> > > Dear reviewer n65V,
> > >
> > > Thank you for taking the time to review our paper and providing valuable feedback. Hopefully you have had an opportunity to read our responses. Please let us know if you have any further questions or comments. We would be happy to answer them. If you found our responses satisfactory, we would appreciate if you consider updating your score.

---

> > > > ### Comment · Reviewer_n65V · 2023-11-23
> > > > **thanks for you response**
> > > >
> > > > Thanks for the detailed explanation which addressed most of my concerns, so I raise my rating to marginally above. I would suggest clarifying the choices for definitions for different biases and keeping them consistent, leading with more informative examples for unfairness rather than trucks and cars, and better motivations for the causal framework in the updated paper, specifically, what other frameworks could one consider, and do they lead to the same theoretical results?

---

> > > > > ### Author Response · Authors · 2023-12-01
> > > > > **Response to Reviewer n65V**
> > > > >
> > > > > Thank you for your comments. We are pleased that our responses have effectively addressed most of your concerns, and we appreciate that you have raised your score. In the Introduction, we have enhanced the illustration of representation harm by incorporating a more informative example from the Wall Street Journal article titled "Google Mistakenly Tags Black People as 'Gorillas'" [1]. To provide further clarity on the choices of different definitions of representation harms in Sections 2.1.2 and 2.2.2 we will elaborate on these aspects in our revised version.
> > > > >
> > > > > We reiterate that causal mediation analysis (CMA) is a principled way to dissect a total effect and tease out the indirect effect that is transmitted through a mediator, which in our case is representation harm. Specifically, in instances where a group is underrepresented (the treatment case), we observe an associated allocation harm in a classification task compared to the control case where the group is adequately represented. Through the application of CMA, we can accurately quantify the degree to which this allocation harm is attributed to representation harm.  A parallel application of CMA is evident in [2], where it is used to understand the roles of individual neurons and attention heads in mediating gender bias within large language models. Furthermore, [4] uses CMA to investigate the impact of debiasing methods on toxicity detection, while [3] employs it to assess the influence of attention heads on in-context learning tasks. In our revision, we shall improve Section 4.1 to better motivate the use of CMA and incorporate these references.
> > > > >
> > > > > Regarding your mention of *other frameworks*, if you are referring to alternative frameworks of causal mediation analysis (CMA), we want to emphasize that in this paper, we have opted for the simplest mediation analysis framework. Our approach considers the entire set of embeddings as a mediator. Although it is conceivable to conduct a more detailed mediation analysis by treating individual embedding components (or even finer details of the feature map) as separate mediators to explore the mechanism through which underrepresentation leads to stereotyping, we want to clarify that such a detailed analysis is beyond the scope of the current study.
> > > > >
> > > > > ---
> > > > > # References
> > > > >
> > > > > [1] Barr, A. (2015). Google mistakenly tags black people as ‘gorillas,’showing limits of algorithms. The Wall Street Journal, 1(7), 2015.
> > > > >
> > > > > [2] Vig, J., Gehrmann, S., Belinkov, Y., Qian, S., Nevo, D., Singer, Y., & Shieber, S. (2020). Investigating gender bias in language models using causal mediation analysis. Advances in neural information processing systems, 33, 12388-12401.
> > > > >
> > > > > [3] Todd, E., Li, M. L., Sharma, A. S., Mueller, A., Wallace, B. C., & Bau, D. (2023). Function vectors in large language models. arXiv preprint arXiv:2310.15213.
> > > > >
> > > > > [4] Jeoung, S., & Diesner, J. (2022). What changed? investigating debiasing methods using causal mediation analysis. arXiv preprint arXiv:2206.00701.

---

### Official Review · Reviewer_X2wa · 2023-11-20

**Soundness:** 2 fair
**Presentation:** 3 good
**Contribution:** 3 good
**Rating:** 5
**Confidence:** 3

**Summary:**

The study presented in this paper concentrates on examining how contrastive learning (CL) can cause representation harm, particularly when certain groups are not adequately represented in the training data. The researchers demonstrate that these underrepresented groups often merge into other semantically similar groups that are better represented. Further, through a theoretical analysis involving graphs, it is shown that increased connectivity between two groups of nodes leads to their convergence. Lastly, a causal analysis reveals that the detrimental effects on representation caused by CL are irreparable for subsequent tasks, even when a probe is trained using the CL representations.

**Strengths:**

1.	The composition of the paper is clear and comprehensive, discussing results in depth and highlighting representational harm. The paper offers practical insights to diminish the adverse effects of CL techniques.
2.	Section 4 presents an intriguing analysis, exploring how representational biases can lead to allocation disparities.
3.	The paper's empirical research is commendable, and its theoretical framework provides a robust underpinning for the empirical observations. The outcomes from both empirical and theoretical perspectives appear promising.

**Weaknesses:**

1.	The text in the figures is too small for easy readability and needs enlargement for better clarity.
2.	A more thorough justification is needed for the chosen metric of representation harm, given its critical importance to the paper's analysis.
3.	The necessity for different data setups in Sections 5 and 6, particularly the choice of causal mediation analysis for Section 6, lacks proper motivation and explanation.
4.	The paper's primary limitation is the absence of a comprehensive large-scale study, especially in the realm of self-supervised learning (SSL).

**Questions:**

See weaknesses above

---

> ### Author Response · Authors · 2023-11-20
> **Response to Reviewer X2wa**
>
> Thank you for your insightful comments and questions. Please see our responses below.
>
>
> **Enlarge texts in figures:** Thank you for the suggestion. We have updated some of the figures in the main draft with a larger text and shall update our other figures (in the main and supplementary draft) shortly. If there are any figures remaining that you find difficult to read, let us know which ones and we will try to improve their readability.
>
> **Justification of the representation harm metric:** We reiterate that the key idea behind measuring representation harm is to measure *collapse* between groups in the data, while relevant measures of collapse, the definitions of groups, and the kinds of groups between which collapse is most problematic may vary across domains and applications.
>
>
> **Different data setups in Sections 5 and 6:** We assume that this was a typo, and you meant Sections 3 and 4. If so, then we point out that the synthetic experiments in Section 3 are used to motivate and interpret our theoretical analysis (Section 3.2) and to draw its connections with our empirical studies in Section 2. On the other hand, the causal mediation analysis in Section 4 shows that in CIFAR10 the downstream allocation harm in the classification task is partly caused by the representation harm.
>
>
> **Motivation for causal framework:** As we explain in the second line of the first paragraph of Section 4 and in Section 4.1, causal mediation analysis [1] is the statistically principled way to dissect total allocation harm (**TE**) and tease out the part caused by representation harm through the natural indirect effect (**NIE**).
>
> **Absence of a comprehensive large-scale study:** We believe that the experiments conducted with the two datasets, the synthetic dataset, and the theoretical analysis were sufficient to demonstrate that under-representation can lead to representation and allocation harm in contrastive learning settings. In addition to various CL methods, such as SimCLR, SimSiam, and SimCSE, we have also included the *boosted contrastive learning* method [2] (in Section 5) to further strengthen our experimental results. Although we acknowledge that additional experiments would be beneficial, the current paper already consists of nine pages of main text and 11 pages of supplementary material, not including references.
>
> ---
> # References
>
> [1] Pearl, J. (2022). Direct and indirect effects. In Probabilistic and causal inference: the works of Judea Pearl (pp. 373-392).
>
> [2] Zhou, Z., Yao, J., Wang, Y. F., Han, B., & Zhang, Y. (2022, June). Contrastive learning with boosted memorization. In International Conference on Machine Learning (pp. 27367-27377). PMLR.

---

### Meta-Review · Area_Chair_g1xp · 2023-12-12

**Metareview:**

This paper addresses the representation harm in contrastive learning (CL) caused by group underrepresentation, offering both empirical and theoretical insights. It is well-written and provides a systematic examination using controlled experiments (CIFAR10), natural imbalances (BiasBios), and artificial graphs. The study sensibly links representation harm to resultant allocation harms and examines the causal relationships involved.

The authors have been proactive in addressing reviewer concerns. They provided clarifications on the metric choices, enhanced readability of figures, incorporated more compelling examples, and justified the use of causal mediation analysis. However, notable concerns include the lack of large-scale studies and the potential insufficiency of the work to overcome representational harm, requesting further exploration in robust CL methods. The criticisms raise essential points on the need for an intuitive definition of harms, consistent metric choices, illustrative examples of real-world implications, and motivation for theoretical frameworks used. While some of these points were addressed satisfactorily, lingering doubts on large-scale applicability and the methodology's limitations suggest room for improvement.

Despite these concerns, the contributions of the paper are clear. It advances the conversation on biases in self-supervised learning by demonstrating empirical harms and their causal pathways. Considering the reviewers' feedback and authors' response, the paper is recommended for acceptance provided that the authors can improve on the minor issues highlighted.

**Justification For Why Not Higher Score:**

The criticisms raise essential points on the need for an intuitive definition of harms, consistent metric choices, illustrative examples of real-world implications, and motivation for theoretical frameworks used. While some of these points were addressed satisfactorily, lingering doubts on large-scale applicability and the methodology's limitations suggest room for improvement.

**Justification For Why Not Lower Score:**

Despite these concerns, the contributions of the paper are clear. It advances the conversation on biases in self-supervised learning by demonstrating empirical harms and their causal pathways. The paper should ensure clarity in defining representation harms consistently across sections and further clarify choices and alternates for the causal framework.

---

### Decision · Program_Chairs · 2024-01-16

Accept (poster)